# Transcriptome Profile of Membrane and Extracellular Matrix Components in Ligament-Fibroblastic Progenitors and Cementoblasts Differentiated from Human Periodontal Ligament Cells

**DOI:** 10.3390/genes13040659

**Published:** 2022-04-08

**Authors:** Seyoung Mun, Seong Min Kim, Min-Jeong Choi, Young-Joo Jang

**Affiliations:** 1Department of Nanobiomedical Science & BK21 FOUR NBM Global Research Center for Regenerative Medicine, Dankook University, Cheonan 31116, Korea; munseyoung@gmail.com (S.M.); holly_luv@naver.com (S.M.K.); hanarhm97@naver.com (M.-J.C.); 2Department of Oral Biochemistry, School of Dentistry, Dankook University, Cheonan 31116, Korea

**Keywords:** human periodontal ligament stem cells, periodontal ligament-fibroblastic progenitors, cementoblasts, RNA sequencing analysis, extracellular matrix molecules, cell surface molecules, stem/progenitor cell niches

## Abstract

Ligament-fibroblastic cells and cementoblasts, two types of progenitor cells that differentiate from periodontal ligament stem cells (hPDLSCs), are responsible for the formation of the adhesive tissues in the tooth root. Since one of the factors that determines the fate of stem cell differentiation is the change in the microenvironment of the stem/progenitor cells, this study attempted to compare and analyze the molecular differences in the membrane and ECM of the two progenitor cells. Single cells derived from hPDLSCs were treated with TGF-β1 and BMP7 to obtain ligament-fibroblastic and cementoblastic cells, respectively. The transcriptome profiles of three independent replicates of each progenitor were evaluated using next-generation sequencing. The representative differentially expressed genes (DEGs) were verified by qRT-PCR, Western blot analysis, and immunohistochemistry. Among a total of 2245 DEGs identified, 142 and 114 DEGs related to ECM and cell membrane molecules were upregulated in ligament-fibroblastic and cementoblast-like cells, respectively. The major types of integrin and cadherin were found to be different between the two progenitor cells. In addition, the representative core proteins for each glycosaminoglycan-specific proteoglycan class were different between the two progenitors. This study provides a detailed understanding of cell–cell and cell–ECM interactions through the specific components of the membrane and ECM for ligament-fibroblastic and cementoblastic differentiation of hPDLSCs.

## 1. Introduction

The periodontal ligament (PDL) is located between the alveolar bone and cementum of the tooth. It forms a part of the connective tissue that anchors the tooth roots into the bone socket. PDL contains multipotent postnatal stem cells and periodontal ligament stem cells (PDLSCs) that can differentiate into multiple progenitors [1,2,3,4]. It has been reported that PDLSCs give rise to cementum and Sharpey’s fibers, the end of the PDL fibers that are embedded in the alveolar bone and cementum. Autologous or allogenic PDLSCs are able to regenerate periodontium by restoring periodontal defects in animals, such as swine [1,5,6,7]. Moreover, patients with periodontal defects regain periodontal attachment with significant pocket reduction when treated with PDL progenitors [8]. These reports suggest that PDLSCs may differentiate into both osteo/cementoblasts and PDL fibroblasts. Under physiological conditions, periodontal tissue predominantly remains in an unmineralized fibrous state, suggesting that the fibroblastic state of the PDL cells is dominant and their osteo/cementogenic differentiation is generally suppressed [9,10,11]. Consequently, because the new cementum on the root surface and the new fibrous attachment is needed to re-establish the periodontal tissue [12], cytodifferentiation of PDLSCs into mineralized cementum and non-mineralized ligament should be well-coordinated.

Key signaling pathways, such as transforming growth factor-beta (TGF-β)/bone morphogenic protein (BMP) and Wnt, may be associated with the survival and differentiation of PDLSCs through cell adhesive mechanisms inside the stem cell niche. TGF-β1 enhances the production of fibrogenic collagens in tendon-derived cells, and its intracellular signaling is upregulated during limb tendon development [13]. In addition, TGF-β1 is highly expressed in the PDL compared with that in cementum and inhibits BMP-2-induced bone formation in mouse PDL cells. The regulatory mechanism of TGF-β1 signaling in fibrogenesis has recently been elucidated, in which TGF-β1 stimulates β-catenin activation and β-catenin is essential for ligament differentiation through a mechanism different from canonical Wnt stimulation [14]. In contrast, BMP-7 induces the expression of cementogenic markers, such as cementum attachment protein (CAP) and cementum protein 1 (CEMP1), in human PDLSCs through a mechanism different from odontogenic differentiation [15,16]. BMP-7 treatment accompanied by the inhibition of TGF-β1 signaling had a synergistic effect on cementoblastic differentiation [14].

The stem cell niche plays a role in the regulation of stem cell proliferation and differentiation through physical cell adhesion [17,18]. Cell–cell or cell–ECM interactions occurring inside the niche are mediated by various molecules. The main components of the ECM are structural proteins, glycoproteins, and proteoglycans, which interact with cell surface receptors, mainly integrins [19,20]. Cell adhesion molecules (CAMs) are also involved in stem cell adhesion within the niche to promote the stemness and self-renewal capacity of stem cells [21]. The stem cell niche is a model of cellular adhesion via numerous mechanisms. Although the specialized environment that surrounds stem cells is crucial for the modulation of both fibroblastic and cementoblastic differentiation potentials, the molecular profiles of ECM and cell adhesion molecules that distinguish these two progenitors in hPDLSCs remain unclear and need to be elucidated. In this study, we developed ligament-fibroblastic progenitors and cementoblast-like cells from hPDLSCs as we previously reported [14], and revealed cell surface and ECM molecules that were specifically expressed in these two progenitor cells based on high-throughput RNA-sequencing analysis. Finally, our results have helped elucidate the specific stem cell niches that form different parts of the human periodontium, such as cementum and periodontal ligament.

## 2. Materials and Methods

### 2.1. Cell Culture and Treatments

Adult third molars were collected from patients between the ages of 19 and 23 visiting Yonsei Wooil Dental Hospital and DKU Dental Hospital under guidelines approved by the IRB of the Dankook University (DKU NON2020-008). Periodontal ligament tissues were separated from the surface of the tooth root, and enzymatically digested with 3 mg/mL collagenase (Millipore, Burlington, MA, USA) and 4 mg/mL dispase (Sigma, St. Louis, Mo, USA) at 37 °C for 1 h. Single cell suspension was incubated with α-MEM (Hyclone, Marlborough, MA, USA) containing 20% fetal bovine serum (Hyclone, Marlborough, MA, USA) and antibiotics (Lonza, Basel Switzerland) at 37 °C in 5% CO_2_. Each of the three cell batches was not combined and used separately in experiments. hPDLCs were cultured in 6-well plates at a density of 4 × 10^4^ cells per well in α-MEM containing 5% FBS. Cells were treated with the followed cytokines and chemicals for cytodifferentiation [14]: 10 ng/mL TGF-β1 (Sino Biological, Wayne, PA, USA) for ligament-fibroblastic differentiation; 10 μM SB431542 (TOCRIS, Abington, UK) and 100 ng/mL BMP-7 (Prospec, Rehovot, Israel) for cementoblastic differentiation.

### 2.2. Flow Cytometry

Cells were detached by enzyme-free dissociation buffer (Millipore, Burlington, MA, USA), and 1 × 10^6^ cells were incubated with 2.5 μg of the purified monoclonal antibody in PBS containing 1% BSA for 1 h on ice. Then, FITC-conjugated anti-mouse IgG (Santa Cruz Biotechnology, Dallas, TX, USA) was added as the secondary antibody for 1 h on ice. After washing, cells were analyzed by FACSCalibur™ (BD Biosciences, Franklin Lakes, NJ, USA). Antibody binding affinity was analyzed using Cell Quest and the WinMDI program ver.2.8 (The Scripps Research Institute, La Jolla, CA, USA).

### 2.3. Quantitative Reverse Transcriptional PCR (qRT-PCR)

cDNA was synthesized from total RNA using the ReverTra Ace™ qPCR RT kit (Toyobo Corporation, Osaka, Japan). The qRT-PCR was performed using iTaq™ Universal SYBR™ Green Supermix (Bio-Rad, Hercules, CA, USA) in a StepOnePlus™ Real-Time PCR System (Thermo Fisher Scientific, Waltham, MA, USA). The primers used for qRT-PCR are shown in Table 1. The cycling parameters of qPCR were as follows: 1 cycle for 1 min at 95 °C, 40 cycles for 15 s at 95 °C, and 1 min at 55–60 °C. During PCR, a dissociation curve was constructed in the range of 65 to 95 °C. GAPDH was used as an internal control to normalize the variability in target gene expression. Student’s *t*-test was applied for statistical analysis. For all graphs, data are represented as mean ± SD and considered statistically significant for a *p*-value less than 0.05.

### 2.4. RNA Extraction and RNA-Seq Library Construction

Cells were harvested and homogenized in 500 ul of TRIzol reagent (Invitrogen, Carlsbad, CA) using a micro-homogenizer following the manufacturer’s instructions. RNA samples were purified using the RNeasy Mini Kit (Qiagen, Germantown, MD, USA). Prior to construction of the RNA-Seq libraries, the quality of all RNA samples was checked by the 28S/18S ratio and RNA integrity number (RIN) value using an Agilent TapeStation 2100 system (Agilent Technologies, Santa Clara, CA, USA). All RNA samples showed RNA integrity number values of more than 7.5 (Appendix A). mRNA molecules were purified from 2 μg of the qualified RNA samples using oligo-dT magnetic beads. cDNA was immediately synthesized by SuperScript III reverse transcriptase (Thermo Fisher Scientific). According to the instructions of the NEBNext^®^ Ultra™ RNA Library Prep Kit (Illumina, San Diego, CA, USA), a sequential process of end-repair, poly-A addition, and adaptor ligation on both ends was carried out. The final selected libraries were evaluated with an Agilent TapeStation 2100 400–500 bp in size. cDNA libraries were sequenced with an Illumina Novaseq 6000 (Illumina, San Diego, CA, USA), which generated paired-end reads of approximately 150 bp in size.

### 2.5. Data Analysis

*Quality control:* Raw sequencing data were evaluated to discard low-quality reads by FAST-QC (https://www.bioinformatics.babraham.ac.uk/ accessed on 11 March 2021) as follows: reads including more than 10% of skipped bases (marked as ‘N’s); sequencing reads including more than 40% of bases whose quality score is less than 20. Quality distributions of nucleotides, GC contents, the proportions of PCR duplication, and k-mer frequencies of sequencing data were also calculated [22].

*Read mapping and differentially expressed genes (DEGs) analysis:* Only highly qualified reads were mapped to the human reference genome (*Homo sapiences*: GRCh38) using the aligner HISAT2 v2.1.0 [23]. We only used uniquely mapped read pairs for the downstream differentially expressed genes (DEGs) analysis. The gene expression level was quantified by DESeq2 v1.26.0 [24]. DEGs for three groups were analyzed using the DESeq2 methods. The DEGs with log_2_ fold-change (log_2_FC) more than 2 and adjusted *p*-value (Q-value) less than 0.01 were considered statistically significant. The overall expression pattern between samples was shown in pairwise correlation analysis and scatterplot, the hierarchical samples clustering heatmaps, and principal components analysis (PCA) plots using ggplot2 R package. The heatmap clustering analysis of DEGs was performed based on the log_2_ FPKM values, and the heat map was generated using hclust2 package (ver.3.6.2, available at https://github.com/SegataLab/hclust2 accessed on 11 March 2021) with the popular clustering distance (*euclidean*) and hierarchical clustering method (*complete*) functions.

### 2.6. GO Functional Enrichment and Protein–Protein Interaction (PPI) Analysis for DEGs

GO and enrichment analysis of DEGs were performed using the Metascape (http://metascape.org/gp/index.html accessed on 11 March 2021). The multiple gene lists identified from DEG analysis were used as the input gene, and three main categories of gene functions (cellular component; CC, molecular function; MF, biological process; BP) were extracted for GO annotation. Functional enrichment analysis was performed with default parameters (min overlap of three, enrichment factor of 1.5, and *p*-value of 0.01) for filtering [25]. The molecular complex detection (MCODE) algorithm was used to identify a densely connected network of PPI. Data integration was achieved through solid statistical procedures and was then visually examined within the PPI network [26].

### 2.7. Western Blot Analysis

Cells were lysed by treatment with 1% NP-40 buffer (20 mM Tris-HCl, pH 8.0, 100 mM NaCl, 2 mM EDTA, pH 8.0, 2 mM EGTA, pH 8.0, 1% NP-40, protease inhibitors). Cell extract was separated on SDS-PAGE, transferred to a PVDF membrane (Millipore, Burlington, MA, USA), and incubated with the primary antibody. After probing with HRP-secondary antibody (GE Healthcare, Chicago, IL, USA), signals were visualized using an ECL Western Blotting Detection Kit (GE healthcare, Chicago, IL, USA) under X-ray film.

### 2.8. Immunohistochemistry

Tissue was embedded in paraffin block and cut into 5~6-μm-thick sections. Endogenous peroxidase activity was inhibited by incubation with 0.3% H_2_O_2_ in PBS for 30 min. The sections were incubated at RT for 1 h in blocking solution (5% horse serum in 0.1% PBST, 0.1% Tween 20 in PBS) and treated with the antibody at 4 °C for 16 h. Then, tissues were washed with 0.1% PBST and incubated with biotin-conjugated anti-mouse IgG (Vector Laboratories, Burlingame, CA, USA) at RT for 1 h. After washing, tissue sections were incubated with VECTASTAIN ABC Reagent (Vector Laboratories, Burlingame, CA, USA) at RT for 30 min and incubated with the DAB substrate for the development of signals. The nucleus was detected by hematoxylin and eosin staining. Tissue on the microscope slides was detected by an Upright FL microscope, Nikon Eclipse 80i (Nikon, Melville, NY, USA).

### 2.9. Antibody Information

The information of the primary antibodies used in this study is as follows: Anti-ADAM12 (sc-293225), anti-alkaline phosphatase (sc-271431), anti-Ang-4 (sc-377497), anti-BCAM (sc-365191), anti-Calpain 5 (sc-271271), anti-cathepsin K (sc-48353), anti-CD109 (sc-271085), anti-CHRDL1 (sc-100333), anti-Crossveinless-2 (sc-377502), anti-CTHRC1 (sc-293270), anti-desmoplakin I/II (sc-390975), anti-EP3 (sc-57105), anti-EphA2 (sc-398832), anti-EphA4 (sc-365503), anti-FEZ1 (sc-393768), anti-FGFR-3 (sc-13121), anti-fibromodulin (sc-166406), anti-glypican-3 (sc-65443), anti-GPNMB (sc-271415), anti-IFITM1/2/3 (sc-374026), anti-IL-6Rα (sc-373708), anti-Integrin α3 (sc-374242), anti-Integrin α8 (sc-365798), anti-Integrin α11 (sc-390091), anti-Integrin β8 (sc-514150), anti-Jagged1 (sc-390177), anti-LTBP-1 (sc-271140), anti-LTBP-2 (sc-166199), anti-M-CSF (sc-365779), anti-MFAP4 (sc-398438), anti-MK (sc-46701), anti-NOPE (IGDCC4, sc-398452), anti-P2X7 (sc-514962), anti-PAI-1 (SERPINE1, sc-5297), anti-PMEPA1 (sc-293372), anti-podoplanin (sc-376695), anti-PPARγ (sc-7273), anti-PTH1R (sc-12722), anti-PXDN (sc-293408), anti-SIRP-α (sc-376884), anti-Tenascin-C (sc-25328), anti-tetranectin (CLEC3B, sc-376940), anti-TM4SF4 (sc-293348), anti-Vasorin (sc-517034), and anti-versican (sc-47769) antibodies were obtained from Santa Cruz Biotechnology (Dallas, TX, USA). Anti-GPM6B (TA361526) and anti-LRRC17 (TA339973) antibodies were obtained from OriGene (Rockville, MD, USA). Anti-LRRC15 (ab157484) and anti-SCGF (CLEC11A, ab90238) antibodies were obtained from Abcam (Waltham, MA, USA).

## 3. Results

### 3.1. Cytodifferentiation of hPDLCs into Ligament-Fibroblastic Cells and Cementoblast-like Cells

To establish PDL-fibroblasts and cementoblast-like cells, we first isolated primary PDL cells from PDL tissues of three independent donors. Primary hPDLCs grew out from the PDL tissues under continuous culture in media containing 20% fetal bovine serum (FBS). After the first passage, hPDLCs were sorted for further differentiation experiments. The entire culture and differentiation procedure is shown in Figure 1.

hPDLCs isolated from 3 independent teeth (#139, #155, and # 159) were verified by the expression of representative mesenchymal stem cell markers (Figure 2A). CD44, CD90, and CD146 proteins are general surface marker proteins of the mesenchymal stem cells isolated from multiple adult and fetal organs, and their expression may be linked to multipotency [27,28,29,30]. These markers were highly expressed in primary hPDLCs (Figure 2A(b–d)). The CD34 protein is a member of a family of transmembrane sialomucin proteins that are expressed in early hematopoietic tissues [31]. Vascular cell adhesion molecule (VCAM)-1 is present on activated endothelial cells, macrophages, and dendritic cells. E-cadherin is expressed in epithelial tissues and is an indicator of mesenchymal to epithelial reverting transitions [31,32]. As expected, these three markers were not expressed in the early passages of hPDLCs (Figure 2A(a,e,f)). The data indicate that hPDLCs repeatedly cultured from different teeth appear to consistently maintain mesodermal stem cell properties. Each cell batch of the three technical replicates were independently used for cytodifferentiation.

In our previous report, we developed induction conditions for ligament-fibroblastic and cementoblastic differentiation from hPDLCs [14]. Low concentrations of TGF-β1 significantly increased the expression of ligament-fibroblast markers. BMP-7 treatment accompanied by the inhibition of TGF-β1 signaling had a synergistic effect on inducing cementoblastic differentiation. By applying differentiation conditions, we investigated the differences in the transcriptional expression of known markers during differentiation of the same stem cell into two different progenitor cells by qRT-PCR. When cells were treated with 10 ng/mL of TGF-β1 for 9 days, the gene expression of the ligament-fibroblastic markers, such as scleraxia (SCX), periodontal ligament-associated protein-1 (PLAP-1), and osteopontin (OPN), were highly increased (Figure 2B, TGF in a–c). The increase in OPN expression was consistent with a previous report that OPN is required for the differentiation and activity of myofibroblasts formed in response to profibrotic TGF-β1 [33]. When TGF-β1 signaling was blocked by treatment with SB431542, a TGF-β type 1 receptor inhibitor, the expression of fibroblastic markers was decreased (Figure 2B, SB in a–c). To induce cementum differentiation in hPDLCs, cells were treated with BMP-7, a potent bone-inducing factor [14,34,35]. Under normal culture conditions, even without the addition of TGF-β1, hPDLCs grew predominantly in a fibroblastic form. SB431542 was treated with BMP7 to completely eliminate the basal profibrotic effect of TGF-β1, which may exist in trace amounts in the FBS medium [14]. Treatment with SB431542 affected the expression of the two representative cementoblastic proteins, except ostericx (OSX) (Figure 2B, SB in d–f). However, when hPDLCs were co-treated with BMP-7 and SB431542, all cementoblastic markers were highly expressed (Figure 2B, SB/BMP7 in d–f). According to previous reports, OSX plays an essential role in cementogenesis of mesenchymal-derived PDL progenitor cells, and SCX counteracts the osteogenic activity regulated by OSX in the PDL [36,37]; thus, the opposite expression patterns of these 2 genes are consistent with those reported in previous results (Figure 2B(a,d)). hPDLCs cultured from three independent tooth samples were separately differentiated into ligament-fibroblastic cells and cementoblast-like cells, and their transcript changes were analyzed in three repetitive attempts by RNA-Seq analysis.

### 3.2. RNA Sequencing and Identification of Differentially Expressed Genes (DEGs)

To explore the gene expression changes, the transcriptomes of three replicates of SB+BMP7-treated cementoblastic cells and TGF-β1-treated fibroblasts were analyzed by RNA sequencing. We first confirmed that the sequencing quality was acceptable by evaluating several quality measures (e.g., total reads and genome mapping coverage). An average of 45.7 million raw reads were produced with a read length of 150 bp. At least 6.75 GB of clean data, accounting for more than 96.87% of the raw data, were prepared for further analysis. A total of 96.83% of the clean data were uniquely mapped to the human reference genome GRCh38 at an average rate of 86% (Appendix A). After gene annotation using the Ensembl database (release 98), 25,133 genes were detected in at least 1 group, and 19,351 genes were commonly expressed in all the groups. The distribution of the genes expressed in the three trial cases showed a consistent expression level, indicating that there were no issues in sample preparation and data production. The box plot denotes statistical values, such as the mean or median, and variations in each sample (Appendix A). To emphasize the association between samples in each group, we confirmed the reproducibility of technical replicates using scatterplot and pairwise correlation analysis based on the overall gene expression in the sample. The results showed that there was a solid similar distribution within the group, and distinct differences were confirmed between the groups (Appendix A). The log_2_-transformed RNA expression value for each sample was calculated to determine the similarity between samples. This was translated into a principal component analysis (PCA) plot, which describes the degree of variability of the entire dataset. The results showed that the development of treatment strategies clearly separated each group. The reproducibility and concordant transcriptome alterations between replicates are shown in Appendix A. Moreover, the hierarchical clustering heatmap also showed transcriptional concordance among the three replicates (Appendix A).

Next, we analyzed DEGs to identify the biological differences between the SB431542 and BMP-7 co-treatment and TGF-β1 treatment on the respective cells (adjusted *p*-value < 0.01 and log_2_FC ≥ 2). In the comparison analysis, we identified 2245 DEGs (865 genes upregulated in SB+BMP7-treated cells and 1380 genes upregulated in TGF-β1-treated cells) as shown in Figure 3A. The data for the normalized gene expression of 2245 DEGs are listed in Appendix A. The PCA and hierarchical clustering heatmap results showed high similarities among the replicates of the independent treatment conditions. Moreover, PCA clearly separated SB+BMP7-treated cells from TGF-β1-treated cells, indicating that the transcriptomic alteration is treatment dependent (Figure 3B–D).

### 3.3. Functional Classification of DEGs

In total, 2245 DEGs with physiologically characterized products were classified into functional categories using the GO database in the Metascape webtool. As for the cellular component (CC), 323 and 266 upregulated genes in ligament-fibroblastic cells and cementoblastic cells, respectively, were significantly associated with 20 GO terms (Figure 4A(a)). In particular, the ECM-related functional group (GO: 0031012) was significantly enriched, with highest number of DEGs. In addition, significant changes in the following function-related genes, such as cell–cell contact zone (GO:0044291) and cell–cell junction (GO:0005911), which determine the fate of cells, were observed. The GO category of the anchored component of the membrane (GO:0031225) was specifically enriched with 86 upregulated DEGs in both progenitors. The gene expression patterns and the informatic details of the representative GO in the CC class are shown in Figure 4A (Figure 4A(b)) and Appendix A.

In biological process (BP) function prediction, 323 and 266 upregulated genes in ligament-fibroblastic cells and cementoblastic cells, respectively, were significantly associated with 20 GO terms, including ECM organization, cell junction organization, and regulation of ossification (Figure 4B(a)). As for the upregulated genes in SB+BMP7-treated cementoblastic cells, the ossification (GO:0001503) and regulation of ossification (GO:0030198) showed high-density enrichment. ECM organization (GO:0001568), muscle structure development (GO:0061061), cell junction (GO:0034330), and transmembrane receptor signaling pathway (GO:0007169) were more closely related to upregulated genes in TGF-β1-treated fibroblastic cells. The gene expression patterns and informatic details of the representative GO in the BP class are shown in Figure 4B (Figure 4B(b)) and Appendix A.

A total of 290 and 228 genes were upregulated in TGF-β1-treated and SB+BMP7-treated cells, respectively, and they were implicated in molecular function (MF) prediction (Figure 4C(a)). TGF-β1-treated fibroblastic cells appear to have induced transcriptomic changes in various genes related to ECM configuration, including the ECM structural constituent (GO:0005201) and CAM binding (GO:0050839). DEGs classified as signaling receptor regulators (GO:0030545) were dominantly screened in SB+BMP7-treated cementoblastic cells. The gene expression patterns and informatic details of the representative GO in the MF class are shown in Figure 4C (Figure 4C(b)) and Appendix A. In addition, the heparin-binding domain-related function, mainly known as a major factor in hard tissue differentiation, such as glycosaminoglycan (GAG) binding (GO:0005539) and ECM binding (GO:0050840), was also dominant in SB+BMP7-treated cementoblastic cells.

### 3.4. Integrative Mining of Biological Pathways Based on Protein–Protein Interaction (PPI)

To explore the networks of the biological pathways among GO categories, PPI analysis was performed using Metascape and Cytoscape software (Figure 5A). The gene group of the MAPK cascade pathway was correlated with the regulation of signaling receptor regulators and epithelial cell proliferation. As expected, the gene groups related to GAG binging and ECM organization interacted closely. The gene group in tissue morphology is closely related to the genes involved in muscle development, ossification, and connective tissue development.

The MCODE enrichment analysis based on PPI analysis resulted in a network characterized by the presence of PPI modules of upregulated DEGs in TGF-β1-treated fibroblastic cells and SB+BMP7-treated cementoblastic cells. To identify a densely connected network of PPI among DEGs upregulated in ligament-fibroblastic cells, seven representative MCODEs enriched in ECM and membrane molecules, cell adhesion, and cell signaling were analyzed. COL10A1 (log_2_FC = 9.29), ITGA11 (log_2_FC = 4.07), LTBP2 (log_2_FC = 3.27), WNT7B (log_2_FC = 7.45), DSP (log_2_FC = 2.64), CACNA2D2 (log_2_FC = 2.58), and ADAMTS8 (log_2_FC = 7.45) were identified as seed protein molecules for meaningful connections for translational activation in TGF-β1-treated fibroblastic cells (Figure 5B and Appendix A).

As for the densely connected network of PPI with upregulated DEGs in cementoblastic cells, six representative MCODEs enriched in CAM, cell signaling, anchored molecules, MAPK cell signaling, and tissue morphogenesis were analyzed. For this, CCR1 (log_2_FC = 8.14), ADRB2 (log_2_FC = 5.46), CHRDL1 (log_2_FC = 6.26), ITGB8 (log_2_FC = 5.3), LY6K (log_2_FC = 4.26), and FZD5 (log_2_FC = 4.09) were defined as seed proteins that could make meaningful connections for translational activation in SB+BMP7-treated cementoblastic cells (Figure 5C and Appendix A). Interestingly, the positive regulation of the MAPK cascade (GO:0043410) was selected as the representative module for the upregulated DEGs. The MAPK signaling pathway is known to be important in promoting the odontogenesis of dental pulp cells. Odontoblastic differentiation by minor trioxide aggregate (MTA) via the MAPK pathway has been clearly demonstrated in human dental pulp stem cells [38,39]. The particulars of PPI modules, including the “phospholipase C-activating G protein-coupled receptor (GPCR) signaling pathway (GO:0007200)” and the “GPCR activity “GO:0008227”, were also very specific to the MAPK signaling pathway. As expected, integrated analysis showed that MAPK pathway regulation and activation of GPCR signaling were detected in the differentiation of cementoblasts upon inhibition of the TGF-β1 signaling pathway [40,41,42].

### 3.5. Expression of DEGs Related to Membrane and ECM Molecules Upregulated during Differentiation Were Verified by qRT-PCR and Protein Analysis

Among the closely related GO groups in Figure 5A, 16 groups related to ECM, cell adhesion, cell–cell junction, and membrane components were selected for further verification studies. Among the 960 genes belonging to 16 groups, 142 genes upregulated by TGF-β1 treatment (Table 2) and 114 genes upregulated by SB+BMP7 treatment (Table 3) were selected after sorting out genes overlapping among the GO groups.

The transcriptional expression of representative ECM and cell surface molecule-related genes upregulated in TGF-β1-treated and SB+BMP7-treated cells, respectively, was verified by qRT-PCR. When 86 of 142 upregulated genes in TGF-β1-treated cells were randomly selected and verified by qRT-PCR, the expression of most of the selected genes was consistent with the RNA-Seq results (Figure 6A).

The expression of 69 upregulated genes in the SB+BMP7-treated cells was also consistent with the RNA-Seq results (Figure 6B).

To verify the RNA-Seq data of genes at the protein level, we performed immunoblot analysis of total cell extracts prepared from hPDLCs treated with SB+BMP7 and TGF-β1, using as many antibodies as possible.

As expected, the protein levels of known protein markers of fibroblastic PDL cells, such as SCX and PLAP-1, were clearly increased in cells treated with TGF-β1 (Figure 7A(a)). In addition to these two markers, OPN/SPP1 expression was increased in ligament-fibroblastic cells differentiated by TGF-β1 treatment. This was consistent with previous findings that OPN is required for the differentiation and activation of myofibroblasts formed in response to the profibrotic cytokine TGF-β1 [33]. The 21 genes transcriptionally upregulated in TGF-β1-treated fibroblastic cells were confirmed to be upregulated at the protein level (Figure 7A(b)). Cementoblast-specific markers, such as CAP, CEMP-1, and OSX, were highly increased in cells treated with SB+BMP7 compared to in those treated with TGF-β1 (Figure 7B(a)). The 28 genes transcriptionally upregulated in cementoblastic cells by SB+BMP7 treatment were also upregulated at the protein level (Figure 7B(b)).

To verify the RNA-Seq results in vivo, immunohistochemistry (IHC) was performed on adult tooth tissue slices using 11 of the antibodies used in the Western blot experiment. Genes for tissue blot analysis were preferentially selected when appropriate antibodies were available for IHC. As mentioned in the materials and methods, paraffin blocks of young adult teeth were prepared to investigate whether the candidate gene products accumulated in the periodontal ligament and cementum. In addition to PLAP-1 and SCX, the proteins encoded by genes verified by qRT-PCR and Western blot analysis, such as CTHRC1, EphA2, ENPP1, FAP-α, and PDPN, accumulated in the periodontal ligament region, which is the outer layer of the tooth root in contact with the alveolar bone (Figure 8A(b,c)). In the tooth slices, cementum was detected in between dentin and ligament, and it was stained with the specific markers CEMP1 and OSX (Figure 8B(b)). As expected, randomly selected gene products, such as ACAN, CD109, CLEC3B, CTSK, ITG-α8, and LGALS3, accumulated in the cementum region (Figure 8B(c)). Based on these experimental results, it was possible to distinguish between ECM and cell surface markers expressed specifically in PDL fibroblastic and cementoblastic differentiation.

## 4. Discussion

The periodontal ligament contains a stem cell population that can be differentiated into progenitor cells with completely different characteristics: fibroblastic PDL progenitors and cementoblastic cells. The purpose of this study was to understand the cellular response to specific environmental factors during ligament and cementoblastic differentiation of hPDLCs. We compared two progenitor cell-specific cell surface factors that can communicate with the environment during the differentiation process.

In previous studies, we established the conditions for differentiation of hPDLCs into ligament-fibroblastic progenitors and cementoblast-like cells [9,14]. Progenitors were obtained from hPDLCs, and transcriptome analysis was performed for high-throughput screening of the cell surface and ECM molecules specifically expressed in both progenitor cells. Low concentrations of TGF-β1 induced fibroblastic cell growth in hPDLCs and upregulated PLAP-1 and OPN. PLAP-1, also known as asporin (ASPN), is a new member of the family of small leucin-rich proteoglycans (SLRPs), and a negative regulator of periodontal ligament mineralization [10,43,44]. OPN, called secreted phosphoprotein 1 (SPP1), is a long-known ECM molecule that is upregulated in fibroblastic proliferation by TGF-β1 and is required for myofibroblast differentiation [33,45]. These two genes stand out in the results as the upregulated DEGs (PLAP-1, log_2_FC = 6.03; OPN, log_2_FC = 5.11) in ligament-fibroblastic progenitors (Table 2 and Figure 6A, Figure 7A and Figure 8A). For cementoblastic differentiation in hPDLSCs, TGF-β1 signaling for the ligament-forming pathway should be completely blocked and induction of the hard-tissue-forming pathway by BMP-7 is required [14]. A representative cementoblastic marker, CEMP-1, was found to be one of the upregulated DEGs, with a log_2_FC value of 6.03, in cementoblastic cells (Table 3 and Figure 6B, Figure 7B and Figure 8B). Cementum-attachment protein (CAP) was not shown as an upregulated DEG in the RNA-Seq results. Cell surface markers specific to cementoblastic cells were evaluated by qRT-PCR, Western blotting, and IHC (Figure 2B, Figure 6B and Figure 7B). The results suggest that the RNA-Seq analysis performed in this study is a reliable method for large-scale analysis of the gene specific to the two progenitor cells.

Among the 18,502 genes that were analyzed using RNA-Seq, 2245 (12.1%) genes were differentially expressed by a factor of 2 or more between the fibroblastic PDL progenitor and cementoblastic cells. RNA-Seq and GO analyses showed that genes upregulated in both types of differentiation were closely related to the functional features of their respective progenitors. In ligament-fibroblastic progenitors induced by TGF-β1, genes related to collagen formation, muscle/ligament differentiation, transmembrane receptor signaling, ECM organization, cell adhesion, and gated ion channels were highly increased (Figure 4A–C). In contrast, genes involved in ossification, membrane anchoring, and GAG/ECM binding were increased in the cementoblastic cells (Figure 4A–C).

It is noteworthy that the types of structural proteins constituting the ECM are significantly different in the two progenitor cells. In ligament-fibroblastic cells, various collagen subtypes, such as COL4A2, COL5A1, COL5A2, COL6A3, COL7A1, COL8A2, COL10A1, COL11A1, COL12A1, COL13A1, COL15A1, COL16A1, COL24A1, and COL26A1, were upregulated. In addition to collagen, CTHRC1 was also upregulated in PDL progenitors. This gene is known as a secreted glycoprotein and has been reported to regulate collagen deposition [46] (Table 2). Only COL14A1 is predominantly upregulated in cementoblastic cells (Table 3).

Laminin is a cell adhesion glycoprotein that is essential for tight binding of cells through interaction with surface receptors on the basement membrane [47]. Based on the RNA-Seq results, it could be predicted that the ECM structure of cementoblastic progenitor cells was mainly composed of laminins, such as LAMA2, LAMA4, LAMA5, LAMB1, LAMB3, and LAMB4, which are responsible for epithelial cell adhesion to the basement membrane and the stability of the ECM [48] (Table 3). Since the types of structural proteins constituting the ECM were different in the two progenitors, it was natural that the expression patterns of the integrins binding to them were different.

Integrins play an important role in defining and shaping the stem cell microenvironment, known as the stem cell niche. Approximately 20 αβ heterodimeric members of integrin mediate the specialized cell–cell and cell–ECM interactions to regulate stem cell functions [49,50]. ITGA4, ITGA10, ITGA11, ITGB1, and ITGBL1 were predominantly expressed in ligament-fibroblastic progenitors, and ITGA2, ITGA8, and ITGB8 were upregulated in cementoblastic cells (Table 2 and Table 3 and Figure 6, Figure 7 and Figure 8). In addition to integrins, genes encoding CAM exhibit different expression patterns. Interestingly, cadherin molecules, such as CDH2, CDH4, CDH11, CDH13, CDH20, CDHR1, and CDHR2, were upregulated in ligament-fibroblastic cells. CADM4 (cell adhesion molecule 4) and MADCAM1 (mucosal vascular addressin cell adhesion molecule 1) were also upregulated following TGF-β1 treatment (Table 2). However, the CDHR3 gene was only upregulated in cementoblasts (Table 3). It has been reported that cells containing a specific cadherin subtype cluster together both in vitro and during development [51,52]. In particular, the cadherin family maintains cell–cell contact by the formation of adherens junctions in a calcium-dependent manner [53], suggesting that these molecules may be essential for the formation of ligaments, which create tension by forming tight bonds between cells.

One of the important results of this study is that the representative core protein types of each proteoglycan (PG) class are different between the two progenitors. PGs are classified according to the GAG type. Neurocan (log_2_FC = 3.16) and aggrecan (log_2_FC = 2.46), belonging to the hyalectan class, were significantly upregulated in ligament-fibroblastic progenitors and cementoblasts, respectively. GPC-3 (log_2_FC = 4.46) in the heparan sulfate PG class was upregulated in cementoblasts. Moreover, osteomodulin (log_2_FC = 3.54) and fibromodulin (log_2_FC = 4.57) in the keratan sulfate PG class were upregulated in fibroblastic and cementoblastic cells, respectively (Table 2 and Table 3). In Table 4, the representative genes belonging to ECM molecules, proteoglycans, and cell adhesion molecules in both progenitors are summarized. These results suggest that the GAG and PGs of the ECM can determine the differentiation fate of hPDLSCs.

Upregulation of the genes encoding extracellular leucine-rich repeat family, such as LRIG1, LRRC15, LRRC17, and LRRC4B, was found only in fibroblastic differentiation induced by TGF-β1 treatment. Among them, LRIG1 has been reported as a cell quiescence factor that negatively regulates mitogenic signals [54], and LRRC15 expression is induced by TGF-β1 in activated fibroblasts of mesenchymal stem cells [55].

Ligands for members of the frizzled receptor family, such as WNT2, WNT3A, WNT4, WNT5B, WNT7B, and WNT9A, were upregulated only in ligament-fibroblastic cells and not in cementoblastic cells. In a previous report, the canonical Wnt pathway potently stimulated fibroblastic activation in tissue fibrosis. Inhibition of canonical Wnt signaling reduces the profibrotic effects of TGF-β1, demonstrating that the interaction between the canonical Wnt pathway and TGF-β1 plays a key role in fibroblastic PDL differentiation [14,56].

Unlike TGF-β1-treated ligament-fibroblastic cells, genes encoding lectins, such as CLEC3B, CLEC11A, LGALS3, and LGALS3BP, were specifically upregulated in cementoblastic cells. Lectins are a class of proteins that can either be free or linked to cell surfaces, and are involved in cell–cell interactions, signaling pathways, and cell development [57]. It is not clear why the expression of these genes is involved in cementoblastic differentiation, but a link can be found in previous studies on CLEC11A function in hard tissue formation. CLEC11A promoted osteogenesis, and CLEC11A-deficient mice showed reduced bone strength and delayed fracture healing [58].

The seven mammalian ENPP proteins, ENPP1-7, which are membrane-bound glycoproteins that hydrolyze extracellular nucleotide triphosphates to produce pyrophosphate, have distinct substrate specificities and participate in different biological processes. According to the RNA-Seq results, ENPP1 (log_2_FC = 3.50) and ENPP2 (log_2_FC = 5.37) were upregulated in fibroblastic and cementoblastic progenitors, respectively (Table 2 and Table 3). ENPP1 negatively regulates bone mineralization by hydrolyzing extracellular nucleoside triphosphates (NTPs) to produce pyrophosphate (PPi), and mutations in this gene result in ectopic ossification of the ligaments [59,60]. The extracellular regions adjacent to the transmembrane domain are disordered and do not interact with the catalytic domain in ENPP1, unlike ENPP2, suggesting that Enpp1 and Enpp2 have distinct roles in the developing periodontium. These results suggest that ENPP modulation should be considered as a potential target for the reconstruction of periodontal tissues [61].

Furthermore, we examined DEGs in ligament-fibroblastic progenitor and cementoblastic cells originating from hPDLSCs. Since the fate of stem cells is determined by the environment around the cells, that is, the niche, the various ECM and cell adhesive molecules presented in this study will serve as useful information to understand the differentiation and regeneration of periodontal ligament tissues.

## 5. Conclusions

PDL-fibroblasts and cementoblasts, two types of progenitor cells that differentiate from hPDLSCs, are responsible for the formation of tissues of the tooth root. Since the fate of stem cells is determined by the environment around the cells, that is, the niche, various ECM and cell adhesive molecules were analyzed through next-generation sequencing and verified by qRT-PCR, Western blot analysis, and immunohistochemistry. The major ECM structural proteins predominantly consisted of various types of collagen and laminin in the ligament-fibroblastic cells and in the cementoblastic cells, respectively. The major types of integrin and cadherin were also found to be different between the two progenitor cells and the representative core proteins for each proteoglycan class were also different between the two progenitors. These results suggest that cell–cell and cell–ECM interactions are important in determining the differentiation and regeneration of periodontal ligament tissues.

## Figures and Tables

**Figure 1 genes-13-00659-f001:**
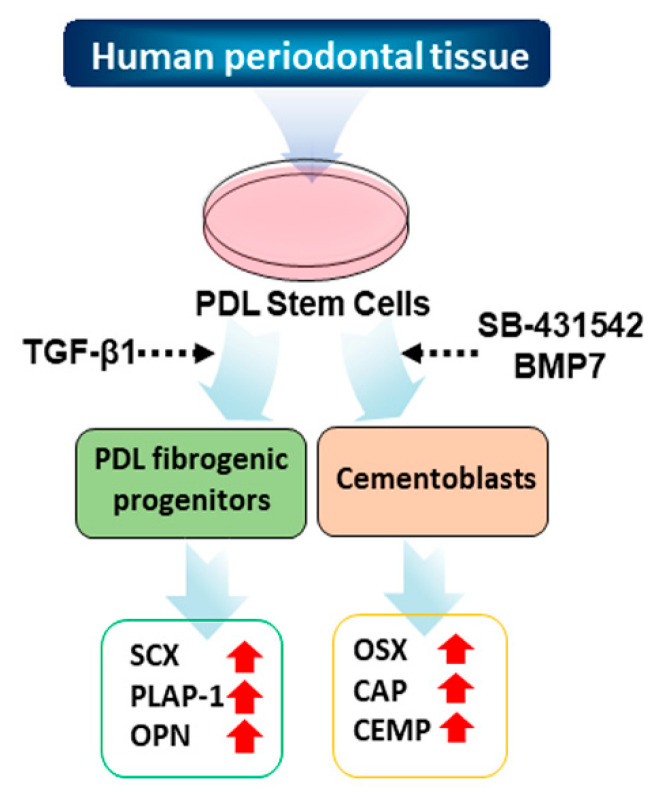
Schematic strategy of the cytodifferentiation of hPDLCs into ligament-fibroblastic progenitors and cementoblast-like cells. The detailed scheme of the differentiation is described in the materials and methods.

**Figure 2 genes-13-00659-f002:**
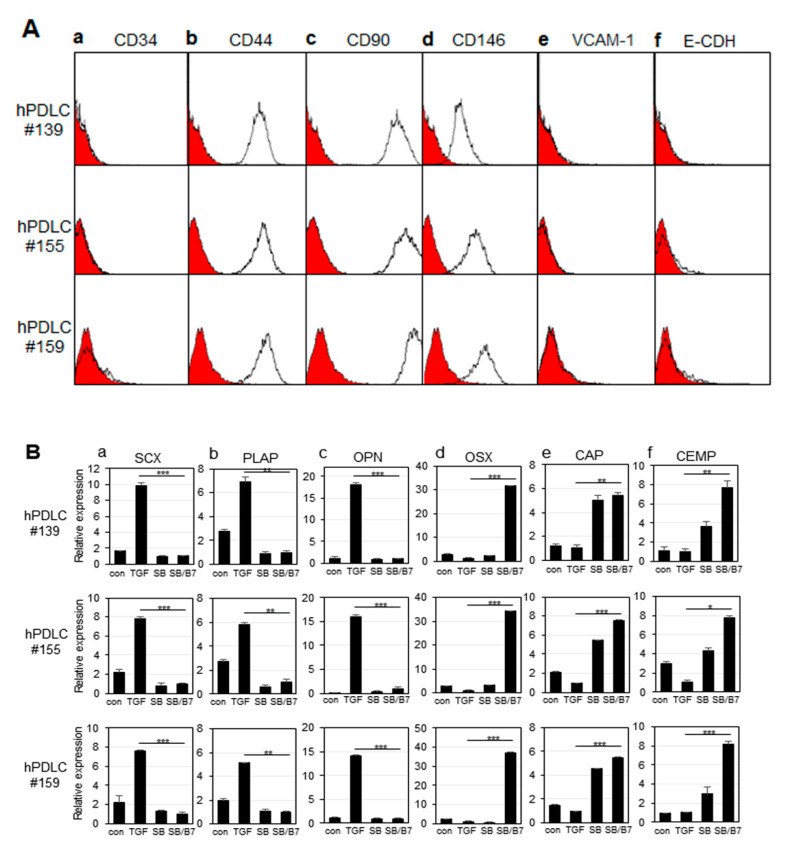
Characterization of hPDLCs and progenitors. The primary cells were independently cultured from tooth root obtained from 3 different patients (numbered as 139, 155, and 159). (**A**) Immunophenotyping of hPDLCs using hematopoietic, mesenchymal, endothelial, and epithelial stem cell markers. Intact cells harvested by non-trypsin methods were incubated with the primary antibodies, and the expression of cell surface antigens was analyzed by FACS as described in the materials and methods. (**a**), FACS histogram on the expression of CD34; (**b**–**d**), FACS histograms on CD44, CD90, and CD146 expressions as mesenchymal stem cell markers; (**e**), FACS histogram on VCAM-1 expression; (**f**), FACS histogram on E-cadherin expression. The curves filled in red were the results of FACS analysis obtained from cells incubated only with FITC-conjugated anti-mouse IgG. (**B**) Relative mRNA expression of fibroblastic and cementoblastic markers in hPDLCs treated with cytokines. mRNA expression was analyzed by qRT-PCR as described in the materials and methods. (**a**–**c**), expressions of ligament-fibroblastic markers SCX, PLAP-1, and OPN, respectively; (**d**–**f**), expressions of cementoblastic markers OSX, CAP, and CEMP-1, respectively. con, no treatment; TGF, TGF-β1 treatment; SB, SB431542 treatment; SB/B7, co-treatment with SB431542 and BMP7. The data were obtained from the average value of 3 individual experiments. Statistical significance of * *p* < 0.1, ** *p* < 0.05, or *** *p* < 0.01 was determined by Student *t*-test.

**Figure 3 genes-13-00659-f003:**
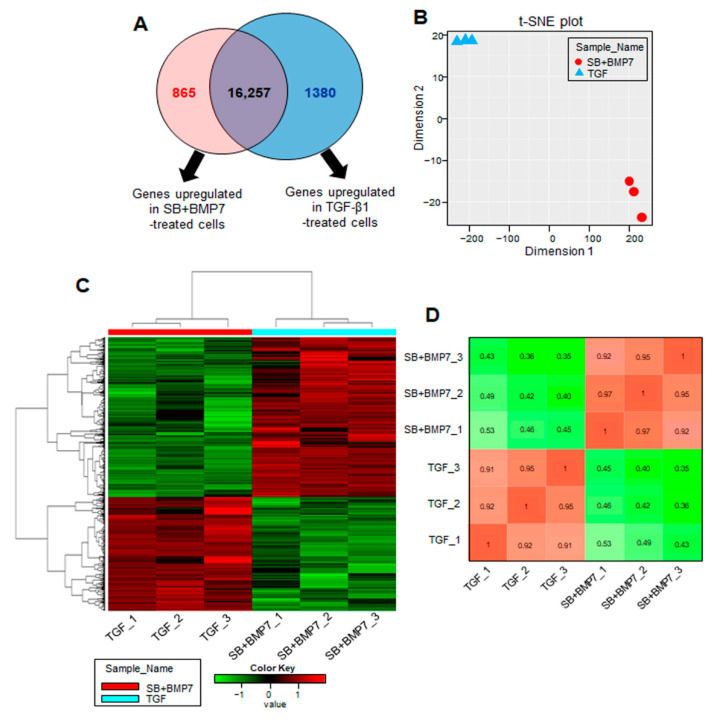
Transcriptome analysis of hPDLCs treated with TGF-β1 and co-treated with SB431542 and BMP7. (**A**) The number of upregulated genes identified in the comparison set, SB+BMP7 vs. TGF-β1. Overlapping areas in the Venn diagram represent genes common to all comparison groups. The size of the circle is proportional to the number of DEGs in the comparison set. (**B**) The MDS plot shows how similar and close the transcriptome changes of each sample are based on log_2_ FPKM values. (**C**) Hierarchical clustering heatmap for 2245 DEGs (865 DEGs upregulated in SB+BMP7-treated cells and 1380 DEGs upregulated in TGF- β1-treated cells) are represented. There is a histogram in the color key showing the number of expression values within each color bar. (**D**) The Euclidean distances were calculated between each sample and the colors indicate the distances. The gradient from red (high) to green (low) indicates relative similarity of gene expression between the samples.

**Figure 4 genes-13-00659-f004:**
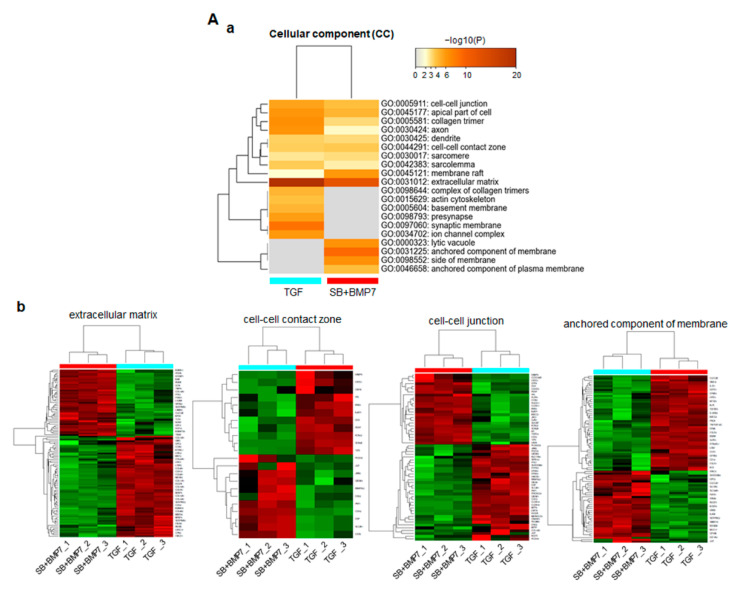
Function prediction analysis for DEGs in TGF-β1-treated and SB+BMP7-treated hPDLCs using Metascape software. Bar chart of the clustered enrichment ontology categories (GO), in which a discrete color scale represents statistical significance. The function classification of upregulated genes in TGF-β1- and SB+BMP7-treated hPDLCs is visualized and screened as follows: cellular component (CC) (**A**), biological process (BP) (**B**), and molecular function (MF) (**C**), respectively. (**a**), the function classification for upregulated genes. The heatmap cells are colored by their p-values within the hypergeometric *p*-value color scale; (**b**), gene expression patterns. Each gene expression across the all samples is colored based on z-score. Red is maximum and green represents minimum expression levels of the samples. The detailed results for the function classification and DEGs are listed in Appendix A.

**Figure 5 genes-13-00659-f005:**
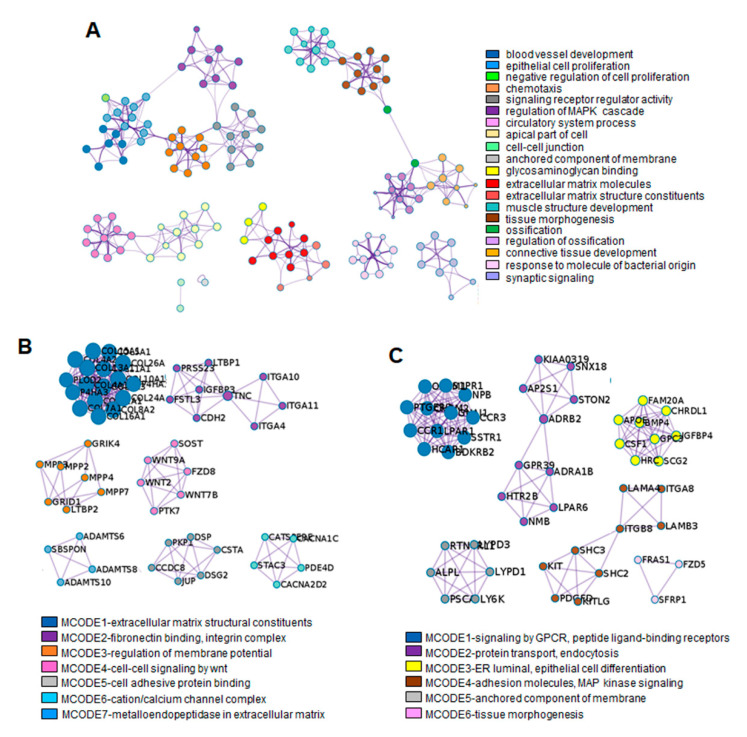
PPI network analysis using Metascape software. (**A**) The enrichment GO clusters including 2245 DEGs. The MCODE algorithm was applied for clustering ontology enrichment to identify neighborhoods. The proteins corresponding with the representative upregulated DEGs in TGF-β1-treated fibroblastic progenitors (**B**) and in SB+BMP7-treated cementoblastic cells (**C**) were densely connected. Detailed information of the MCODE modules is listed in the Appendix A. Each term is represented by a circle node, where its size is proportional to the number of input genes falling into that term, and its color represents its cluster identity. Each cluster identity or MCODE network was assigned a unique color.

**Figure 6 genes-13-00659-f006:**
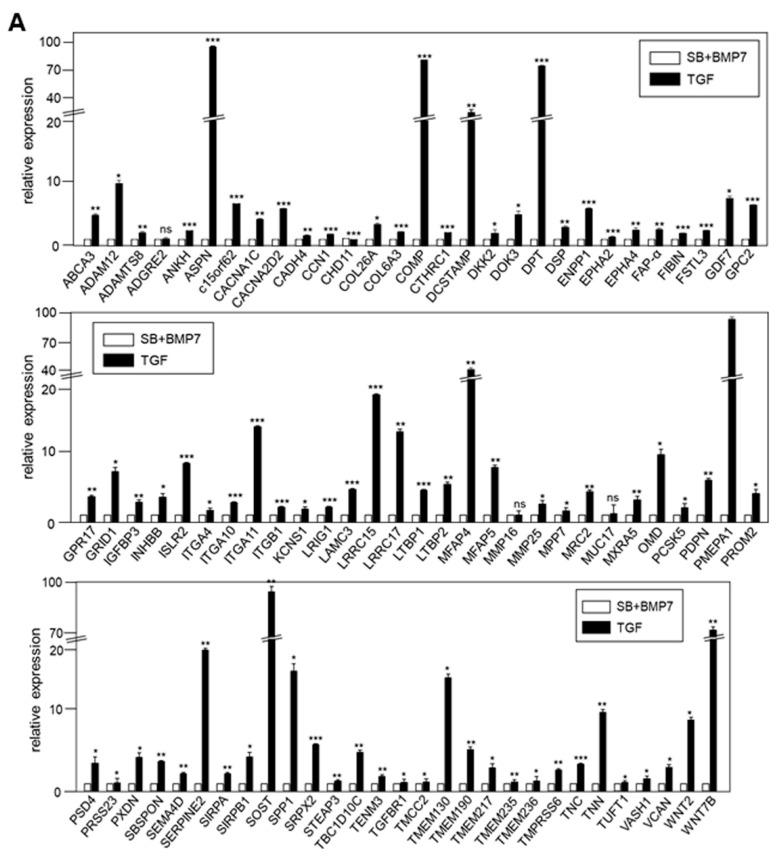
Validation of the expression of DEGs by qRT-PCR. (**A**) In total, 86 of 142 DEGs upregulated by TGF-β1 treatment were confirmed by qRT-PCR. (**B**) In total, 69 of 114 DEGs upregulated by SB431542 and BMP7 treatment were confirmed by qRT-PCR. Statistical significance of * *p* < 0.1, ** *p* < 0.05, or *** *p* < 0.01 was determined by Student *t*-test. ns: not significant.

**Figure 7 genes-13-00659-f007:**
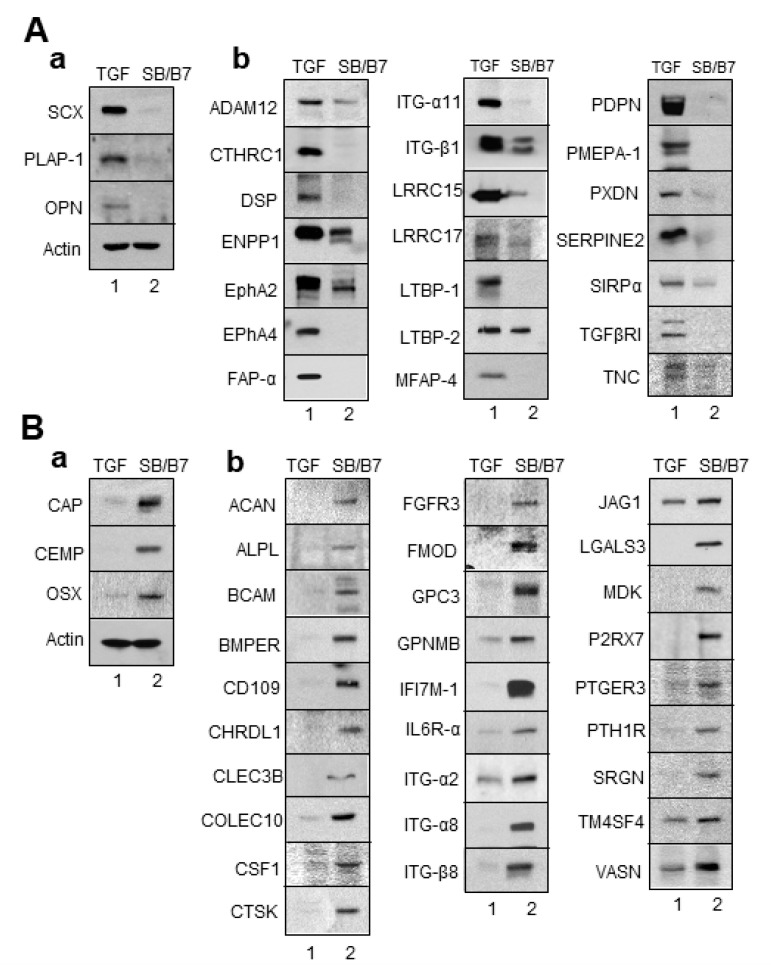
Validation of the expression of DEGs by Western blot analysis. Total cell extracts were obtained from progenitor cells and immunoblot analysis was performed with the primary antibodies. (**A**) The selected genes upregulated by TGF-β1 treatment were confirmed by Western blot analysis. (**B**) The selected genes upregulated by SB431542 and BMP7 treatment were confirmed by Western blot analysis. (**a**), immunoblots of the already known markers; (**b**), immunoblots of the selected upregulated DEGs. Lane 1, cell extract of hPDLCs treated with TGF-β1; lane 2, cell extract of hPDLCs treated with SB431542 and BMP7.

**Figure 8 genes-13-00659-f008:**
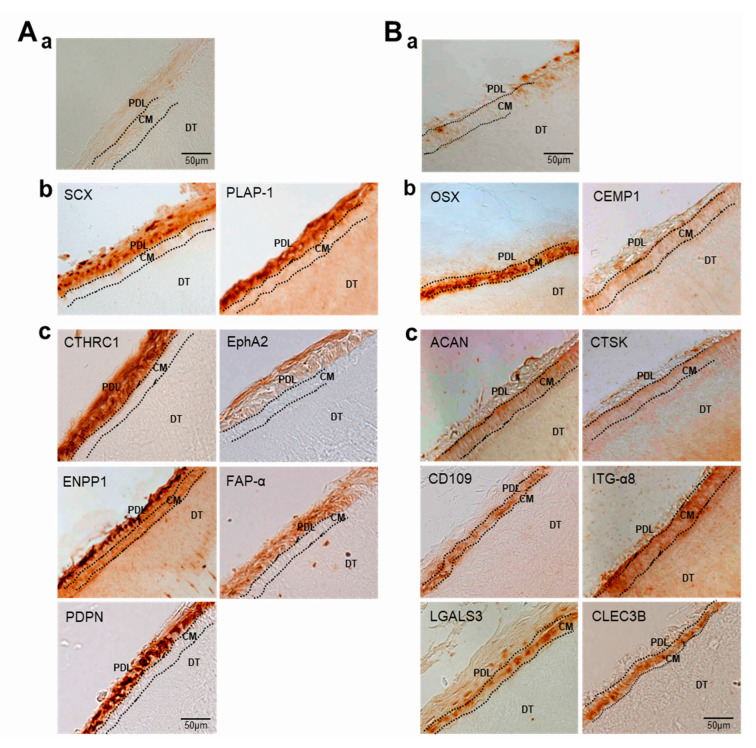
Validation of the expression of DEGs by immunohistochemistry. Immunohistochemical staining of PDL (**A**) and cementum (**B**) in tooth roots. (**a**), negative control by staining with the secondary antibody only; (**b**), staining with the antibodies of the already known markers; (**c**), staining with the antibodies of the selected upregulated DEGs. PDL, periodontal ligament; CM, cementum; DT, dentin.

**Table 1 genes-13-00659-t001:** Primers used for the quantitative real-time-PCR (qPCR).

Gene		Primer Sequence
*Cementum protein 1 (CEMP1)*	Forward Reverse	5′-GATCAGCATCCTGCTCATGTT-3′ 5′-AGCCAAATGACCCTTCCATTC-3′
*Cementum attachment protein (CAP)*	Forward Reverse	5′-TCCAGACATTTGCCTTGCTT-3′ 5′-TTACAGCAATAGAAAAACAGCAT-3′
*Scleraxis (SCX)*	Forward Reverse	5′-AGAAAGTTGAGCAAGGACC-3′ 5′-CTGTCTGTACGTCCGTCT-3′
*Periodontal ligament-associated protein-1* *(PLAP-1)*	Forward Reverse	5′-TTGACCTCAGTCCCAACCAA-3′ 5′-TCGTTAGCTTGTTGTTGTTCAG-3′
*Osterix (OSX)*	Forward Reverse	5′-GAAGGGAGTGGTGGAGCCAAAC-3′ 5′-ATTAGGGCAGTCGCAGGAGGAG-3′
*Osteopontin (OPN)*	Forward Reverse	5′-GTGGGAAGGACAGTTATGAA-3′ 5′-CTGACTTTGGAAAGTTCCTG-3′
*GAPDH*	Forward Reverse	5′-GTATGACAACAGCCTCAAGAT-3′ 5′-CCTTCCACGATACCAAAGTT-3′

**Table 2 genes-13-00659-t002:** The 142 DEGs in GO categories of ECM and membrane molecules were upregulated by TGF-β1 treatment.

	Gene	Gene Description	Expression FPKM						log_2_FC	*p*-Value
			TGF_1	TGF_2	TGF_3	SB + BMP7 _1	SB + BMP7 _2	SB + BMP7 _3		
1	*ABCA3*	ATP binding cassette subfamily A3	271.81	382.68	584.43	88.18	60.93	60.27	2.57	1.20 × 10^−9^
2	*ADAM12*	ADAM metallopeptidase domain 12	13,478.31	15,874.44	15,422.11	3602.47	2575.41	2318.91	2.41	4.65 × 10^−15^
3	*ADAMTS8*	ADAM metallopeptidasethrombospondin	52.47	44.28	11.76	0	0	0	7.45	1.07 × 10^−6^
4	*ADGRE2*	adhesion G protein-coupled receptor E2	25.19	2.11	79.96	2.18	4.13	0	4.05	1.53 × 10^−3^
5	*ANKH*	ANKH inorganic pyrophosphate transport	3257.55	2371.99	2099.01	880.75	700.13	708.77	2.00	2.83 × 10^−8^
6	*ASPN*	asporin	2402.23	1518.08	899.57	10.89	35.11	28.82	6.03	8.42 x 10^−37^
7	*C15orf62*	chromosome 15 open reading frame 62	106	85.39	59.97	31.57	14.46	13.1	2.09	6.49 × 10^−4^
8	*CACNA1C*	calcium voltage-gated channel alpha1 C	294.9	175	95.25	43.55	47.5	27.51	2.26	1.88 × 10^−5^
9	*CACNA2D2*	calcium voltage-gated channel α2δ2	157.42	92.77	336.31	17.42	34.08	48.47	2.58	7.48 × 10^−6^
10	*CADM4*	cell adhesion molecule 4	99.7	339.46	279.87	90.36	23.75	66.82	2.00	4.51 × 10^−4^
11	*CCN1*	cellular communication network factor 1	28,795.29	94,228.27	82,932.51	30,376.6	29,237.16	21,458.45	1.35	1.10 × 10^−3^
12	*CDH11*	cadherin 11	14,571.86	11,323.37	6964.94	5021.03	6761.73	4751.8	2.00	4.18 × 10^−3^
13	*CDH13*	cadherin 13	8984.49	12,760.28	16,782.64	1985.77	1384.77	1008.79	3.14	1.19 × 10^−17^
14	*CDH2*	cadherin 2	6200.25	7637.82	7708.12	414.79	525.61	503.09	3.91	4.96 × 10^−42^
15	*CDH20*	cadherin 20	31.48	31.63	18.81	2.18	8.26	2.62	2.64	7.26 × 10^−3^
16	*CDH4*	cadherin 4	219.34	104.37	130.53	32.66	5.16	7.86	3.31	6.90 × 10^−7^
17	*CDHR1*	cadherin-related family member 1	14.69	12.65	77.61	2.18	2.07	7.86	3.17	2.85 × 10^−3^
18	*CDHR2*	cadherin-related family member 2	39.88	35.84	51.74	8.71	10.33	23.58	2.00	2.64 × 10^−2^
19	*COL10A1*	collagen type X alpha 1 chain	1355.91	2166.42	642.05	2.18	4.13	0	9.29	2.66 × 10^−35^
20	*COL11A1*	collagen type XI alpha 1 chain	29,096.49	51,801.2	30,256.27	362.53	270.55	175.56	7.10	2.75 × 10^−80^
21	*COL12A1*	collagen type XII alpha 1 chain	66,629.64	178,722.94	70,352.58	19,720.51	16,547.03	11,341.71	2.73	5.33 × 10^−11^
22	*COL13A1*	collagen type XIII alpha 1 chain	297	382.68	1407.57	148.06	156.96	129.7	2.28	1.32 × 10^−5^
23	*COL15A1*	collagen type XV alpha 1 chain	12,370.07	11,658.62	32,661.01	111.05	74.35	90.4	7.70	2.95 × 10^−72^
24	*COL16A1*	collagen type XVI alpha 1 chain	31,832.45	27,283.21	31,856.69	6276.29	4930.86	6681.61	2.36	1.07 × 10^−16^
25	*COL24A1*	collagen type XXIV alpha 1 chain	109.14	273.04	168.16	0	8.26	0	6.03	1.15 × 10^−4^
26	*COL26A1*	collagen type XXVI alpha 1 chain	31.48	7.38	9.41	0	0	0	6.29	4.66 × 10^- 4^
27	*COL4A2*	collagen type IV alpha 2 chain	203,405.68	125,145.38	272,092.39	43,512.72	23,430.62	22,768.57	2.75	1.73 × 10^−12^
28	*COL5A1*	collagen type V alpha 1 chain	24,9774.3	332,644.3	360,388.25	49,241.4	27,740.86	17,513.68	3.32	3.84 × 10^−17^
29	*COL5A2*	collagen type V alpha 2 chain	55,067.66	97,987.62	61,875.42	20,400.94	22,349.44	15,365.08	2.00	1.40 × 10^−8^
30	*COL6A3*	collagen type VI alpha 3 chain	314,887.46	360,692.87	235,327.44	65,972.38	43,872.77	37,084.25	2.64	6.00 × 10^−15^
31	*COL7A1*	collagen type VII alpha 1 chain	53,082.06	89,729.92	127,376.18	4213.23	3072.11	4547.43	4.53	1.98 × 10^−35^
32	*COL8A2*	collagen type VIII alpha 2 chain	16,937.36	35,120.28	31,222.87	865.51	685.67	509.64	5.34	1.83 × 10^−48^
33	*COMP*	cartilage oligomeric matrix protein	27,604.14	12,697.02	40,509.06	33.75	14.46	23.58	10.15	6.63 × 10^−105^
34	*CTHRC1*	collagen triple helix repeat containing 1	9353.09	17,816.31	12,341.22	2109.88	4094.42	4571.01	1.89	3.83 × 10^−7^
35	*DCSTAMP*	dendrocyte expressed seven ransmem	52.47	8.43	2.35	0	0	0	6.68	2.36 × 10^−4^
36	*DKK2*	dickkopf WNT signaling inhibitor 2	31.48	27.41	16.46	0	2.07	0	5.11	4.38 × 10^−4^
37	*DOK3*	docking protein 3	209.89	184.49	297.51	51.17	35.11	23.58	2.66	1.43 × 10^−8^
38	*DPT*	dermatopontin	2481.99	3934.35	3488.94	8.71	14.46	20.96	7.85	3.05 × 10^−80^
39	*DSP*	desmoplakin	6371.32	10,325.03	11,460.46	1871.46	1403.36	1268.2	2.64	6.29 × 10^−15^
40	*ENPP1*	ectonucleotide pyrophos/diesterase 1	3558.74	1596.09	646.75	226.45	131.15	159.83	3.50	4.37 × 10^−12^
41	*EPHA2*	EPH receptor A2	3040.31	3286	3801.73	858.98	582.41	458.54	2.42	8.73 × 10^−13^
42	*EPHA4*	EPH receptor A4	200.45	266.72	82.31	34.84	36.14	15.72	2.66	2.25 × 10^−6^
43	*FAP*	fibroblast activation protein alpha	9452.55	7967.79	6516.92	2987.36	3330.27	3150.84	1.35	3.30 × 10^−6^
44	*FGFR2*	fibroblast growth factor receptor 2	140.63	516.57	208.14	102.34	107.39	77.3	2.00	1.50 × 10^−3^
45	*FIBIN*	fin bud initiation factor homolog	464.91	288.86	105.83	23.95	17.55	0	4.36	1.18 × 10^−3^
46	*FJX1*	four-jointed box kinase 1	1566.85	1407.38	913.69	265.64	230.28	207	2.48	4.18 × 10^−13^
47	*FSTL3*	follistatin-like 3	15,432.42	22,173.4	31,363.98	2731.52	2565.08	4155.7	2.89	2.40 × 10^−16^
48	*GDF6*	growth differentiation factor 6	686.35	767.47	301.03	8.71	1.03	0	7.45	7.83 × 10^−27^
49	*GDF7*	growth differentiation factor 7	58.77	10.54	77.61	4.35	2.07	2.62	4.04	4.14 × 10^−5^
50	*GP1BA*	glycoprotein Ib platelet subunit alpha	75.56	71.69	49.39	9.8	5.16	15.72	2.72	7.02 × 10^−5^
51	*GPC2*	glypican 2	96.55	107.53	127	21.77	15.49	0	3.13	2.37 × 10^−5^
52	*GPR17*	G protein-coupled receptor 17	153.22	72.74	142.29	22.86	6.2	5.24	3.42	2.79 × 10^−7^
53	*GPR173*	G protein-coupled receptor 173	1388.45	1055.27	1095.95	207.94	95	94.33	3.16	1.78 × 10^−15^
54	*GPR3*	G protein-coupled receptor 3	107.05	73.8	81.14	27.22	8.26	31.44	2.00	1.21 × 10^−3^
55	*GPR65*	G protein-coupled receptor 65	2.1	2.11	7.06	0	0	0	4.18	1.38 × 10^−1^
56	*GPR68*	G protein-coupled receptor 68	771.36	792.77	841.95	160.04	121.85	75.99	2.75	1.30 × 10^−13^
57	*GPR75*	G protein-coupled receptor 75	60.87	72.74	30.57	21.77	10.33	6.55	2.08	4.71 × 10^−3^
58	*GRID1*	glutamate ionotropic receptor delta type 1	50.37	53.77	183.44	13.06	0.00	13.10	3.49	4.52 × 10^−5^
59	*IGDCC4*	immunoglobulin superfamily DCC 4	295.95	345.78	344.54	60.97	67.12	82.54	2.25	1.30 × 10^−9^
60	*IGFBP3*	insulin-like growth factor binding protein	133,088.22	454,889.51	369,842.6	5129.90	4639.65	5426.52	5.99	4.17 × 10^−49^
61	*INHBB*	inhibin subunit beta B	183.66	6061.76	4668.38	222.09	268.49	242.37	3.90	5.42 × 10^−4^
62	*ISLR2*	Immune superfamily leucine rich repeat 2	93.4	8.43	4.7	0	0	0	7.43	1.08 × 10^−5^
63	*ITGA10*	integrin subunit alpha 10	909.89	216.12	131.7	67.5	61.96	57.65	2.77	3.56 × 10^−6^
64	*ITGA11*	integrin subunit alpha 11	88,084.95	38,513.81	59,493.02	7139.62	2383.33	1548.56	4.07	6.58 × 10^−16^
65	*ITGA4*	integrin subunit alpha 4	269.71	242.47	196.38	46.81	53.7	78.61	2.01	1.32 × 10^−6^
66	*ITGB1*	integrin subunit beta 1	44,313.76	77,566.33	43,614.65	30,612.85	29,449.88	13,221.73	1.18	2.93 × 10^−3^
67	*ITGBL1*	integrin subunit beta like 1	26,618.69	30,220.26	36,433.34	4537.66	4131.6	2634.65	3.05	1.13 × 10^−20^
68	*KCNS1*	potassium voltage-gated channel S-1	74.51	298.34	358.65	32.66	22.72	36.68	3.01	1.34 × 10^−7^
69	*LAMC3*	laminin subunit gamma 3	52.47	36.90	524.45	5.44	4.13	5.24	5.39	1.34 × 10^−7^
70	*LMCD1*	LIM and cysteine-rich domains 1	8100.84	3588.56	6799.13	686.96	670.18	775.59	3.14	1.17 × 10^−18^
71	*LRIG1*	leucine-rich and immunoglobulin like 1	92.35	250.9	124.65	39.19	6.2	6.55	3.16	7.05 × 10^−6^
72	*LRP4*	LDL receptor-related protein 4	199.4	107.53	82.31	34.84	34.08	5.24	2.38	2.85 × 10^−4^
73	*LRRC15*	leucine rich repeat containing 15	16,141.86	29,535.02	29,021.56	1241.11	555.56	459.85	5.05	2.74 × 10^−33^
74	*LRRC17*	leucine rich repeat containing 17	375.71	187.65	87.02	19.6	20.65	28.82	3.26	1.14 × 10^−8^
75	*LRRC4B*	leucine rich repeat containing 4B	6.3	68.52	44.68	9.8	2.07	9.17	2.53	1.03 × 10^−2^
76	*LTBP1*	latent TGFβ binding 1	14,018.79	13,426.54	10,422.13	1694	1483.9	1099.19	3.15	8.01 × 10^−24^
77	*LTBP2*	latent TGFβ binding 2	116,984.14	81,316.19	94,494.1	18,144.09	8082.48	4121.64	3.27	1.07 × 10^−12^
78	*MADCAM1*	mucosal vascular addressin adhesion 1	107.05	129.67	156.4	17.42	26.85	45.85	2.16	3.69 × 10^−5^
79	*MATN1*	matrilin 1	97.6	59.04	71.73	21.77	8.26	0	2.91	4.60 × 10^−4^
80	*MFAP4*	microfibril-associated protein 4	55,864.2	17,925.95	20,462.08	1122.44	509.09	411.38	5.54	4.50 × 10^−30^
81	*MFAP5*	microfibril-associated protein 5	2533.41	1892.32	2996.23	361.44	399.63	433.65	2.65	3.13 × 10^−17^
82	*MMP16*	matrix metallopeptidase 16	119.64	113.86	2.35	6.53	14.46	2.62	3.31	9.03 × 10^−4^
83	*MMP25*	matrix metallopeptidase 25	431.33	420.63	579.73	133.91	62.99	79.92	2.38	4.61 × 10^−9^
84	*MPP4*	membrane palmitoylated protein 4	101.80	258.28	116.42	7.62	4.13	5.24	4.82	1.88 × 10^−13^
85	*MPP7*	membrane palmitoylated protein 7	52.47	207.68	131.7	11.98	20.65	55.03	2.20	8.67 × 10^−4^
86	*MRC2*	mannose receptor C type 2	100,932.53	121,686.48	126,924.63	28,287.41	16,143.27	15,961.19	2.54	3.35 × 10^−14^
87	*MUC17*	mucin 17, cell surface associated	8.4	26.36	18.81	0	0	1.31	5.46	1.11 × 10^−3^
88	*MXRA5*	matrix remodeling-associated 5	5479.27	5427.12	5249.28	900.35	471.92	423.17	3.18	9.64 × 10^−19^
89	*NCAN*	neurocan	1.04946746	6.32531874	3.52774206	0	0	1.31011976	3.16	2.23 × 10^−1^
90	*OMD*	osteomodulin	209.89	139.16	28.22	19.6	12.39	0	3.54	3.16 × 10^−5^
91	*PAQR6*	progestin and adipoQ receptor family 6	209.89	158.13	165.8	41.37	25.82	32.75	2.43	1.00 × 10^−7^
92	*PCSK5*	subtilisin/kexin type 5	192.05	208.74	226.95	45.72	32.01	69.44	2.12	1.68 × 10^−6^
93	*PCSK6*	subtilisin/kexin type 6	43.03	40.06	42.33	0	4.13	17.03	2.64	6.78 × 10^−3^
94	*PDPN*	podoplanin	128.04	1922.9	1988.47	3.27	1.03	2.62	9.21	1.57 × 10^−13^
95	*PLOD2*	Procoll-lys2-oxoglutarate 5-dioxygenase 2	10,439.05	29,561.37	19,199.14	1570.97	2468.01	1755.56	3.36	9.51 × 10^−18^
96	*PMEPA1*	prostate transmembrane, androgen 1	12,009.06	13,479.25	18,425.4	124.11	143.54	132.32	6.79	7.97 × 10^−105^
97	*PROM2*	prominin 2	59.82	63.25	62.32	8.71	4.13	13.1	2.88	3.81 × 10^−5^
98	*PRSS23*	serine protease 23	40,093.85	63,023.36	54,863.44	11,832.96	14,735.78	12,987.21	2.01	2.11 × 10^−11^
99	*PSD4*	pleckstrin and Sec7 domain containing 4	59.82	247.74	311.62	32.66	24.78	44.54	2.62	8.19 × 10^−6^
100	*PTPRU*	tyrosine phosphatase receptor type U	1407.34	1327.26	2036.68	421.32	280.88	203.07	2.40	1.44 × 10^−10^
101	*PVRIG*	PVR-related immunoglobulin domain	49.32	41.11	83.49	5.44	19.62	5.24	2.52	1.00 × 10^−3^
102	*PXDN*	peroxidasin	32,845.18	39,677.67	39,607.14	9793.85	6020.3	4295.88	2.48	4.16 × 10^−12^
103	*SBSPON*	somatomedin B and thrombospondin 1	54.57	41.11	75.25	13.06	8.26	7.86	2.56	2.45 × 10^−4^
104	*SEMA4D*	semaphorin 4D	16.79	67.47	96.42	15.24	8.26	0	2.92	1.93 × 10^−3^
105	*SERPINE1*	serpin family E member 1	86,930.54	300,561.22	410,070.62	42,236.78	31,631.85	25,584.02	3.01	9.21 × 10^−11^
106	*SERPINE2*	serpin family E member 2	45,408.36	80,420.1	94,846.87	1113.73	1099.76	1297.02	5.99	1.28 × 10^−70^
107	*SIRPA*	signal regulatory protein alpha	2514.52	3857.39	4502.57	1010.3	628.88	645.89	2.26	1.60 × 10^−10^
108	*SIRPB1*	signal regulatory protein beta 1	4.2	27.41	15.29	0	2.07	0	4.43	8.55 × 10^−3^
109	*SLC1A3*	solute carrier family 1 member 3	612.89	342.62	163.45	90.36	71.25	47.16	2.43	1.60 × 10^−6^
110	*SLC1A4*	solute carrier family 1 member 4	3576.59	1455.88	3318.43	538.9	435.77	423.17	2.59	1.48 × 10^−11^
111	*SOST*	sclerostin	37.78	15,296.72	5796.08	2.17	4.13	7.86	10.57	2.20 × 10^−10^
112	*SPP1*	secreted phosphoprotein 1	73.46	2.11	0	2	0	0	5.11	1.10 × 10^−1^
113	*SRPX2*	sushi repeat containing protein X-linked 2	3357.25	3835.25	4890.63	604.22	582.41	897.43	2.56	1.24 × 10^−15^
114	*STEAP3*	STEAP3 metalloreductase	8126.03	4228.48	3346.65	1465.38	814.75	539.77	2.48	1.89 × 10^−8^
115	*STX1B*	syntaxin 1B	279.16	628.31	408.04	23.95	19.62	37.99	4.04	1.09 × 10^−18^
116	*SUSD4*	sushi domain containing 4	24.14	10.54	7.06	0	0	0	6.07	9.63 × 10^−4^
117	*TBC1D10C*	TBC1 domain family member 10C	94.45	81.17	70.55	2.18	8.26	0	4.51	3.48 × 10^−8^
118	*TENM2*	teneurin transmembrane protein 2	841.67	3346.09	1709.78	366.89	254.03	182.11	2.88	7.89 × 10^−10^
119	*TENM3*	teneurin transmembrane protein 3	3424.41	2982.39	2079.02	682.61	669.15	355.04	2.32	2.77 × 10^−10^
120	*TENM4*	teneurin transmembrane protein 4	1542.72	526.06	1133.58	77.3	46.47	52.4	4.20	2.65 × 10^−21^
121	*TGFBR1*	transforming growth factor beta receptor 1	2238.51	5185.71	3991.05	776.24	1227.81	930.19	2.00	9.98 × 10^−8^
122	*TMC7*	transmembrane channel-like 7	156.37	293.07	172.86	6.53	10.33	9.17	4.59	8.29 × 10^−17^
123	*TMCC2*	transmembrane and coiled-coil domain 2	356.82	426.96	384.52	116.49	100.17	39.3	2.19	8.24 × 10^−7^
124	*TMEM130*	transmembrane protein 130	159.52	71.69	48.21	22.86	22.72	2.62	2.53	7.06 × 10^−4^
125	*TMEM139*	transmembrane protein 139	2.1	4.22	3.53	0	0	0	3.98	1.76 × 10^−1^
126	*TMEM178A*	transmembrane protein 178A	85.01	121.24	99.95	9.8	15.49	11.79	3.05	9.11 × 10^−8^
127	*TMEM190*	transmembrane protein 190	137.48	192.92	183.44	8.71	6.2	5.24	4.67	3.25 × 10^−16^
128	*TMEM217*	transmembrane protein 217	99.7	171.84	105.83	26.13	6.2	18.34	2.91	1.37 × 10^−6^
129	*TMEM235*	transmembrane protein 235	0	4.22	4.1	0	0	0	2.75	4.89 × 10^−1^
130	*TMEM236*	transmembrane protein 236	8.4	14.76	14.11	0	0	0	5.90	1.57 × 10^−3^
131	*TMEM45A*	transmembrane protein 45A	1796.69	4340.22	4396.74	644.5	739.37	851.58	2.25	2.54 × 10^−9^
132	*TMPRSS6*	transmembrane serine protease 6	32.53	108.58	76.43	10.89	2.07	5.24	3.59	6.32 × 10^−6^
133	*TNC*	tenascin C	24,412.71	23,432.14	17,615.19	5277.96	2571.28	1179.11	2.86	4.24 × 10^−10^
134	*TNN*	tenascin N	8.4	6.33	42.33	2.18	0	0	4.68	4.28 × 10^−3^
135	*TUFT1*	tuftelin 1	1971.95	1876.51	1728.59	363.62	326.31	247.61	2.58	1.51 × 10^−16^
136	*VASH1*	vasohibin 1	1080.95	1000.45	1461.66	213.38	157.99	260.71	2.51	2.44 × 10^−13^
137	*VCAN*	versican	11,073.98	8312.52	6030.09	4871.88	4336.06	3145.6	2.07	2.17 × 10^−3^
138	*WNT2*	Wnt family member 2	33.58	238.25	1100.66	4.35	0	2.62	7.64	2.55 × 10^−6^
139	*WNT3A*	Wnt family member 3A	6.3	2.11	7.06	0	1.03	0	3.68	1.13 × 10^−1^
140	*WNT4*	Wnt family member 4	6.3	10.54	55.27	4.35	3.1	0	3.24	1.11 × 10^−2^
141	*WNT7B*	Wnt family member 7B	309.59	63.25	16.46	2.18	0	0	7.45	2.29 × 10^−8^
142	*WNT9A*	Wnt family member 9A	141.68	342.62	318.67	25.04	10.33	13.1	4.05	4.10 × 10^−14^

**Table 3 genes-13-00659-t003:** 114 genes among ECM and cell surface molecules were upregulated by SB431542 and BMP7 treatment.

	Gene	Gene Description	Expression FPKM						log_2_FC	*p*-Value
			TGF_1	TGF_2	TGF_3	SB + BMP7 _1	SB + BMP7 _2	SB + BMP7 _3		
1	*ACAN*	aggrecan	5755.28	19,484.09	9976.45	95,265.73	49,581.21	48,577.93	2.46	3.03 × 10^−8^
2	*ACVRL1*	activin A receptor-like type 1	447.07	86.45	157.57	1108.29	732.14	813.58	2.00	1.58 × 10^−4^
3	*ADGRL2*	adhesion G protein-coupled receptor L2	295.95	496.54	482.12	2503.98	2153.06	2304.50	2.44	4.60 × 10^−14^
4	*ADGRL4*	adhesion G protein-coupled receptor L4	60.87	33.74	18.81	133.91	129.08	254.16	2.16	6.55 × 10^−5^
5	*ADRA1B*	adrenoceptor alpha 1B	107.05	64.31	97.60	521.48	374.85	572.52	2.43	1.87 × 10^−10^
6	*ADRB2*	adrenoceptor beta 2	0.00	6.33	0.00	131.73	55.76	98.26	5.46	3.84 × 10^−9^
7	*ALPL*	alkaline phosphatase	183.66	42.17	23.52	5539.25	3622.50	2811.52	5.57	7.43 × 10^−23^
8	*ANGPT4*	angiopoietin 4	17.84	10.54	9.41	141.53	45.44	82.54	2.82	1.93 × 10^−5^
9	*AP2S1*	adaptor related protein complex 2	6716.59	4073.51	5124.63	9324.62	14,215.33	43241.81	2.03	1.84 × 10^−5^
10	*ATP2B4*	ATPase plasma membrane Ca transport 4	1794.59	4852.57	4029.86	14,412.06	16,734.98	13,110.37	2.04	6.62 × 10^−8^
11	*BCAM*	basal cell adhesion molecule	201.50	140.21	143.46	738.13	631.98	858.13	2.18	1.46 × 10^−10^
12	*BMPER*	BMP binding endothelial regulator	113.34	86.45	109.36	274.35	183.81	123.15	2.03	4.23 × 10^−2^
13	*CCDC173*	coiled-coil domain containing 173	52.47	16.87	38.81	313.54	334.58	356.35	3.20	2.97 × 10^−12^
14	*CCDC85B*	coiled-coil domain containing 85B	3737.15	3738.26	4376.75	11,024.06	19,396.09	65,553.15	2.98	2.13 × 10^−9^
15	*CCR1*	C-C motif chemokine receptor 1	0.00	0.00	0.00	84.92	74.35	10.48	8.14	4.20 × 10^−8^
16	*CD109*	CD109 molecule	1301.34	2117.93	781.98	2820.79	3504.78	3960.49	3.36	7.86 × 10^−4^
17	*CD74*	CD74 molecule	93.40	86.45	95.25	301.57	339.74	305.26	4.93	8.26 × 10^−7^
18	*CDHR3*	cadherin related family member 3	40.93	120.18	129.35	654.30	875.68	1109.67	3.17	4.86 × 10^−13^
19	*CEMP1*	cementum protein 1	529.00	251.00	290.00	659.00	952.00	2268.00	3.92	8.98 × 10^−5^
20	*CHRDL1*	chordin-like 1	2.10	2.11	0.00	145.88	102.23	83.85	6.26	1.21 × 10^−11^
21	*CHRM2*	cholinergic receptor muscarinic 2	0.00	0.00	0.00	546.52	578.28	360.28	11.26	1.74 × 10^−19^
22	*CLEC11A*	C-type lectin domain containing 11A	4940.89	7577.73	8578.29	8711.69	12,058.15	28,970.68	2.83	4.62 × 10^−3^
23	*CLEC3B*	C-type lectin domain family 3 member B	1022.18	934.04	923.09	6228.39	9039.74	23,090.86	3.70	4.33 × 10^−18^
24	*COL14A1*	collagen type XIV alpha 1 chain	940.32	82.23	79.96	18,954.08	16,285.78	11,871.00	5.40	2.45 × 10^−6^
25	*COLEC10*	collectin subfamily member 10	12.59	40.06	42.33	176.37	222.02	404.83	3.06	6.80 × 10^−9^
26	*COLEC11*	collectin subfamily member 11	9.44	2.11	5.88	31.57	17.55	52.40	2.51	6.37 × 10^−3^
27	*CSF1*	colony-stimulating factor 1	885.75	566.12	692.61	6847.85	4825.53	4377.11	2.89	4.62 × 10^−18^
28	*CTSK*	cathepsin K	4442.40	2848.50	1412.27	29,076.71	32,350.57	26,248.25	3.32	5.48 × 10^−17^
29	*CXCL14*	C-X-C motif chemokine ligand 14	0.00	3.16	4.70	291.77	214.79	171.63	6.45	7.27 × 10^−19^
30	*DIO2*	iodothyronine deiodinase 2	8.40	27.41	28.22	108.87	118.75	120.53	2.43	1.89 × 10^−5^
31	*DIO3*	iodothyronine deiodinase 3	2.10	6.33	0.00	50.08	61.96	75.99	4.44	5.31 × 10^−7^
32	*DKK1*	dickkopf WNT signaling inhibitor 1	348.42	128.61	761.99	1695.09	2692.09	2165.63	2.38	1.89 × 10^−6^
33	*ENPP2*	ectonucleotide pyrophos phosphodiesterase	134.33	389.01	30.57	9419.34	7800.57	5771.08	5.37	2.14 × 10^−20^
34	*FAM20A*	FAM20A Golgi-associated secretory pathway	129.08	104.37	83.49	637.97	369.69	434.96	2.17	2.09 × 10^−8^
35	*FGFR3*	fibroblast growth factor receptor 3	30.43	48.49	75.26	209.03	130.11	165.08	3.48	5.01 × 10^−4^
36	*FMOD*	fibromodulin	245.58	265.66	429.21	12,123.64	6428.19	3741.70	4.57	8.73 × 10^−26^
37	*FRAS1*	Fraser extracellular matrix complex 1	197.30	139.16	87.02	799.10	729.04	565.97	2.29	4.71 × 10^−9^
38	*FZD5*	frizzled class receptor 5	4.20	6.33	11.76	149.15	121.85	107.43	4.09	2.13 × 10^−11^
39	*GDF5*	growth differentiation factor 5	11.54	12.65	12.94	393.02	314.96	461.16	4.96	2.24 × 10^−28^
40	*GPC3*	glypican 3	94.45	200.30	65.85	2736.96	2676.60	2590.11	4.46	9.96 × 10^−29^
41	*GPM6B*	glycoprotein M6B	40.93	48.49	42.33	266.73	390.34	227.96	2.74	5.95 × 10^−11^
42	*GPNMB*	glycoprotein nmb	2946.90	1876.51	996.00	29,581.86	25,649.76	24,648.59	3.77	2.64 × 10^−22^
43	*GPR149*	G protein-coupled receptor 149	4.20	0.00	2.35	14.15	8.26	14.41	2.47	8.17 × 10^−2^
44	*GPR150*	G protein-coupled receptor 150	0.00	14.76	21.17	43.55	47.50	28.82	2.00	7.11 × 10^−2^
45	*GPR27*	G protein-coupled receptor 27	0.00	0.00	0.00	6.53	2.07	13.10	5.14	2.38 × 10^−2^
46	*GPR39*	G protein-coupled receptor 39	448.12	99.10	69.38	1376.10	1283.57	1041.55	2.57	3.39 × 10^−6^
47	*GPR78*	G protein-coupled receptor 78	56.67	102.26	42.33	1704.89	1164.82	1011.41	4.26	1.38 × 10^−25^
48	*GPRC5A*	G protein-coupled receptor C-5A	0.00	0.00	27.05	130.64	88.81	144.11	3.73	8.22 × 10^−2^
49	*GPRC5B*	G protein-coupled receptor C-5B	125.94	99.10	109.36	414.79	412.02	427.10	2.00	3.90 × 10^−8^
50	*GPRC5C*	G protein-coupled receptor C-5C	0.00	0.00	0.00	4.35	2.07	7.86	4.54	8.02 × 10^−2^
51	*HTRA4*	HtrA serine peptidase 4	2.10	2.11	0.00	2.18	5.16	24.89	2.87	8.35 × 10^−2^
52	*IFITM3*	interferon-induced transmembrane 3	14,498.39	8629.84	8947.53	34,491.84	37,412.57	45,127.08	2.00	8.77 × 10^−9^
53	*IGSF1*	immunoglobulin superfamily member 1	0.00	0.00	0.00	6.53	12.39	7.86	5.46	8.30 × 10^−3^
54	*IGSF10*	immunoglobulin superfamily member 10	6.30	10.54	7.06	97.98	53.70	31.44	2.94	8.31 × 10^−5^
55	*IL6R*	interleukin 6 receptor	48.28	82.23	61.15	586.80	479.15	377.31	2.91	1.23 × 10^−13^
56	*IL6ST*	interleukin 6 signal transducer	2402.23	1910.25	1358.18	4513.70	6767.93	5055.75	1.51	6.00 × 10^−6^
57	*ITGA2*	integrin subunit alpha 2	113.34	113.86	61.15	243.87	399.63	242.37	2.00	1.76 × 10^−4^
58	*ITGA8*	integrin subunit alpha 8	15.74	240.36	567.97	4677.01	4731.56	4508.12	4.07	3.26 × 10^−4^
59	*ITGB8*	integrin subunit beta 8	16.79	71.69	9.41	769.70	1493.20	1642.89	5.30	4.13 × 10^−21^
60	*JAG1*	jagged canonical Notch ligand 1	737.78	245.63	111.71	2075.04	3094.82	2748.63	2.83	5.50 × 10^−8^
61	*KCNS2*	K voltage-gated channel subfamily S-2	0.00	0.00	0.00	195.96	208.59	276.44	10.12	1.13 × 10^−15^
62	*KIT*	KIT receptor tyrosine kinase	10.49	4.22	4.70	561.76	713.55	330.15	6.35	6.98 × 10^−31^
63	*KITLG*	KIT ligand	130.13	90.66	15.29	696.76	1505.59	1015.34	3.75	1.20 × 10^−11^
64	*LAMA2*	laminin subunit alpha 2	299.10	275.15	298.68	1216.07	675.35	496.54	1.50	2.56 × 10^−4^
65	*LAMA4*	laminin subunit alpha 4	316.94	258.28	575.02	13,448.58	8266.29	5958.43	4.58	6.77 × 10^−29^
66	*LAMA5*	laminin subunit alpha 5	3412.87	1355.73	1654.51	5047.16	4180.13	6347.53	2.00	1.24 × 10^−3^
67	*LAMB1*	laminin subunit beta 1	7315.84	3533.75	3089.13	15,267.77	12,843.99	12,105.51	2.00	5.54 × 10^−5^
68	*LAMB3*	laminin subunit beta 3	110.19	188.71	333.96	2153.43	1341.40	1604.90	3.00	7.09 × 10^−13^
69	*LAMB4*	laminin subunit beta 4	0.00	0.00	0.00	2.18	7.23	3.93	4.45	9.27 × 10^−2^
70	*LGALS3*	galectin 3	4240.90	3437.81	3265.51	11,502.00	11,054.42	8020.55	2.00	1.54 × 10^−6^
71	*LGI4*	leucine-rich repeat LGI family 4	65.07	126.51	122.30	523.66	541.10	955.08	2.67	4.36 × 10^−11^
72	*LRP3*	LDL receptor-related protein 3	2546.01	1470.64	1814.44	9173.29	7222.29	7282.96	2.01	6.31 × 10^−10^
73	*LYPD1*	LY6/PLAUR domain containing 1	396.70	95.93	97.60	1670.05	1442.60	1091.33	2.82	2.58 × 10^−8^
74	*LYPD3*	LY6/PLAUR domain containing 3	2.10	6.33	14.11	48.99	68.15	81.23	3.13	3.37 × 10^−5^
75	*LY6K*	lymphocyte antigen 6 family member K	20.99	40.06	84.67	599.87	772.42	1464.71	4.26	7.84 × 10^−18^
76	*LZTS1*	leucine zipper tumor suppressor 1	548.87	113.86	268.11	1579.69	1216.45	1065.13	2.04	2.21 × 10^−5^
77	*MDK*	midkine	647.52	465.97	312.79	7641.51	10,287.17	19,284.96	4.68	1.94 × 10^−31^
78	*METRN*	meteorin, glial cell differentiation	1373.75	1754.22	2243.64	7122.20	8525.48	16,704.03	2.57	7.84 × 10^−12^
79	*MFRP*	membrane frizzled-related protein	1960.41	1194.43	1541.62	5003.61	5795.18	9632.00	2.10	2.20 × 10^−9^
80	*NIPAL1*	NIPA like domain containing 1	8.40	8.43	2.35	50.08	34.08	45.85	2.74	7.13 × 10^−4^
81	*NMB*	neuromedin B	188.90	120.18	65.85	1245.46	1642.93	2480.06	3.81	1.23 × 10^−19^
82	*P2RX7*	purinergic receptor P2X 7	12.59	51.66	47.04	321.16	217.89	140.18	2.61	2.74 × 10^−6^
83	*PSCA*	prostate stem cell antigen	38.83	8.43	47.04	182.90	198.27	348.49	2.93	2.42 × 10^−7^
84	*PTGER3*	prostaglandin E receptor 3	73.46	225.60	27.05	461.60	544.20	431.03	2.13	1.76 × 10^−4^
85	*PTH1R*	parathyroid hormone 1 receptor	16.79	84.34	174.04	307.01	251.96	441.51	2.00	2.19 × 10^−3^
86	*RSPO2*	R-spondin 2	2.10	0.00	4.70	42.46	36.14	24.89	3.94	2.02 × 10^−4^
87	*RTN4RL1*	reticulon 4 receptor-like 1	3.15	0.00	4.70	578.09	400.66	403.52	7.48	4.94 × 10^−28^
88	*S1PR1*	sphingosine-1-phosphate receptor 1	0.00	0.00	0.00	23.95	40.27	31.44	7.30	1.33 × 10^−6^
89	*SDCBP*	syndecan binding protein	2215.43	2959.19	1688.61	8677.94	14,739.91	12,611.21	2.38	1.88 × 10^−12^
90	*SECTM1*	secreted and transmembrane 1	4.20	2.11	8.23	955.87	651.60	1020.58	7.51	6.72 × 10^−43^
91	*SFRP1*	secreted frizzled-related protein 1	6.30	4.22	2.35	58.79	74.35	62.89	3.90	2.32 × 10^−7^
92	*SMOC2*	SPARC-related modular Ca binding 2	0.00	0.00	0.00	8.71	4.13	0.00	4.43	1.37 × 10^−1^
93	*SPINT2*	serine peptidase inhibitor, Kunitz type 2	1100.89	128.61	636.17	1598.20	2313.12	5435.69	2.29	9.22 × 10^−5^
94	*SRGN*	serglycin	1565.81	880.27	1694.49	2947.08	4715.04	6190.32	2.00	4.79 × 10^−6^
95	*SSTR1*	somatostatin receptor 1	2.10	0.00	0.00	56.61	66.09	39.30	6.22	9.43 × 10^−7^
96	*SUSD2*	sushi domain containing 2	12.59	8.43	2.35	3012.40	1671.85	2088.33	8.15	2.57 × 10^−61^
97	*SUSD3*	sushi domain containing 3	13.64	24.25	11.76	68.59	62.99	83.85	2.10	5.75 × 10^−4^
98	*TGFBR3*	transforming growth factor β receptor 3	50.37	48.49	14.11	206.85	296.37	199.14	2.62	3.07 × 10^−7^
99	*TM4SF1*	transmembrane 4 L six family 1	151.12	59.04	14.11	1907.38	2590.90	3331.63	5.10	5.21 × 10^−20^
100	*TM4SF4*	transmembrane 4 L six family member 4	0.00	0.00	0.00	39.19	68.15	65.51	8.15	4.73 × 10^−9^
101	*TMDD1*	transmembrane and death domain 1	11.54	8.43	11.76	60.97	81.58	87.78	2.84	4.56 × 10^−6^
102	*TMEM144*	transmembrane protein 144	10.49	8.43	16.46	67.50	54.73	104.81	2.66	3.33 × 10^−5^
103	*TMEM154*	transmembrane protein 154	2.10	0.00	0.00	8.71	6.20	15.72	3.81	3.74 × 10^−2^
104	*TMEM229B*	transmembrane protein 229B	4.20	6.33	12.94	197.05	90.87	78.61	3.97	2.78 × 10^−9^
105	*TMEM26*	transmembrane protein 26	8.40	0.00	0.00	411.52	566.92	366.83	7.29	2.27 × 10^−6^
106	*TMEM273*	transmembrane protein 273	0.00	0.00	0.00	19.60	39.24	60.27	7.60	2.98 × 10^−7^
107	*TMEM35A*	transmembrane protein 35A	46.18	13.70	9.41	161.13	161.09	108.74	2.62	2.05 × 10^−5^
108	*TMEM37*	transmembrane protein 37	6.30	2.11	9.41	35.93	25.82	66.82	2.83	9.86 × 10^−4^
109	*TMEM38B*	transmembrane protein 38B	116.49	128.61	125.82	266.73	579.31	656.37	2.00	9.00 × 10^−7^
110	*TMTC1*	Transmem man-transferase cadherin 1	124.89	223.49	175.21	1965.08	2167.51	1822.38	3.50	4.78 × 10^−26^
111	*TXK*	TXK tyrosine kinase	6.30	1.05	0.00	13.06	9.29	22.27	2.56	6.44 × 10^−2^
112	*VASN*	vasorin	55.62	51.66	45.86	118.67	122.88	136.25	2.90	3.76 × 10^−3^
113	*WFDC1*	WAP four-disulfide core domain 1	104.95	37.95	119.94	1775.65	2085.93	3503.26	4.78	5.86 × 10^−28^
114	*XCR1*	X-C motif chemokine receptor 1	0.00	0.00	0.00	8.71	13.42	2.62	5.36	1.31 × 10^−2^

**Table 4 genes-13-00659-t004:** The representative genes in GO categories of ECM and membrane molecules and cell adhesion upregulated in BMP7-induced cementoblastic cells and TGF-β1-induced PDL-fibroblasts.

GO Category	Cementoblast	PDL-Fibroblast
Cell surface ECM Proteoglycan	Aggrecan, Glypican 3 Decorin Fibromodulin TGFBR3/Podocan	Nerocan Biglycan, Asporin Osteomodulin COL15A1
Cell adhesion	Integrin A2 Integrin A8 Integrin B8 CDH10,18,19,R3	Integrin A4 Integrin A10 Integrin A11 Integrin B1 CDH2,4

## Data Availability

The data that support the findings of this study are available from the corresponding author upon reasonable request.

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
