# Peer review of "Transcriptome Profile of Membrane and Extracellular Matrix Components in Ligament-Fibroblastic Progenitors and Cementoblasts Differentiated from Human Periodontal Ligament Cells"

_genes, 2022, doi:10.3390/genes13040659_

Round 1
Reviewer 1 Report
- In Figure 8 the authors tried to validate of expression of DEGs by immunohistochemistry of PDL and cementum in tooth roots to demonstrated that these kinds of proteins were actually found both in the experimental differentiated cells and in fully formed eruption tooth. However, the authors didn’t show the negative control for the immunohistochemistry straining for confirmation on the specificity of the primary antibody and the non-specific binding of the secondary antibody.
- In Figure 8 why did the specific marker of cementoblast (cementum attachment protein; CAP) could not detect by immunohistochemistry in cementum on the root surface.
Author Response
Answers for Reviewer 1’s comments
- In Figure 8 the authors tried to validate of expression of DEGs by immunohistochemistry of PDL and cementum in tooth roots to demonstrated that these kinds of proteins were actually found both in the experimental differentiated cells and in fully formed eruption tooth. However, the authors didn’t show the negative control for the immunohistochemistry straining for confirmation on the specificity of the primary antibody and the non-specific binding of the secondary antibody.
--> As Reviewer 1’s suggestion, the negative control data for the immunohistochemistry straining with the secondary antibody only were added in Figure 8, A & B. The contents of the figure and the text have been corrected. Please see the revised manuscript lines 455~458, 466~468 and figure 8.
- In Figure 8 why did the specific marker of cementoblast (cementum attachment protein; CAP) could not detect by immunohistochemistry in cementum on the root surface.
--> We have already obtained CAP staining data, but we chose the more specific OSX and CEMP staining results, and to balance the number of PDL marker staining results, they are not included in the text. Please see the picture below (Please see the picture in attached file).

Reviewer 2 Report
Overall, the manuscript entitled “Transcriptome profile of membrane and extracellular matrix components in ligament-fibroblastic progenitors and cementoblasts differentiated from human periodontal ligament cells” is very interesting for in-depth study. The manuscript has been very well written. The research data submitted has also answered the problem of the factors that determine the fate of stem cell differentiation: the change in the microenvironment of the stem/progenitor cells. This study attempted to compare and analyze the molecular differences in the membrane and the ECM of the progenitor cells. The approach method has been carried out correctly and is valid. These findings will pave the way for therapeutic strategies through a detailed understanding of cell-to-cell and cell-to-ECM interactions through the specific components of the membrane and ECM for ligament-fibroblastic and cementoblastic differentiation of hPDLSCs.
Some minor notes are:
- The data in Figure 2 are fascinating and clearly illustrate the molecular differences in membrane and ECM of the two progenitor cells. What are the factors causing the different responses in OSX, CAP, and CEMP in hPDLC incubated with SB431542 alone and in combination with BMP-7?
- Transcriptome analysis has been very well done both in the illustrations and explanations in the manuscript. However, there requires a better resolution in Figures 3C and D.
- Figures 4A-C have been well illustrated and interpreted by the author(s) in the manuscript.
- It is recommended that tables 2 and 3 be simplified with their family groups. Meanwhile, each protein's gene description and expression are presented in the data supplement.
Author Response
- The data in Figure 2 are fascinating and clearly illustrate the molecular differences in membrane and ECM of the two progenitor cells. What are the factors causing the different responses in OSX, CAP, and CEMP in hPDLC incubated with SB431542 alone and in combination with BMP-7?
--> Given that periodontal ligament tissue maintains unmineralized fibrous state under physiological conditions, osteogenic/cementogenic differentiation is generally prevented, and ligament fibroblast differentiation usually occurs predominantly in periodontal ligament stem cells (PDLSCs) (Hyun et al., 2017; Yamada et al., 2007). Based on previous histologic data using in vivo models of periodontal development, BMP-7 is expressed during cementogenesis compared to other BMPs (Ripamonti et al., 2005). Additionally, BMP-7 induces the expression of cementogenic markers, such as cementum attachment protein (CAP) and cementum protein 1 (CEMP1), in both normal and immortal periodontal ligament stem cells through a mechanism different from that for osteogenic and odontogenic differentiation (Torii et al., 2015; Torii et al., 2016). BMP-7 reduces the severity of injury after acute and chronic organ failure by counteracting TGF-β1-mediated profibrotic effects (Weiskirchen et al., 2009). Based on these backgrounds, we found that cementoblastic gene expression was completely decreased by TGF-β1, whereas it was significantly increased when cells were treated with BMP-7 and TGF-β type I receptor inhibitor SB431542 (Lim et al., 2020).
- Hyun, S.Y., Lee, J.H., Kang, K.J., and Jang, Y.J. (2017). Effect of FGF-2, TGF-beta-1, and BMPs on Teno/Ligamentogenesis and Osteo/Cementogenesis of Human Periodontal Ligament Stem Cells. Mol Cells 40, 550-557. 10.14348/molcells.2017.0019.
- Lim, J.C., Bae, S.H., Lee, G., Ryu, C.J., and Jang, Y.J. (2020). Activation of beta-catenin by TGF-beta1 promotes ligament-fibroblastic differentiation and inhibits cementoblastic differentiation of human periodontal ligament cells. Stem Cells. 10.1002/stem.3275.
- Ripamonti, U., Herbst, N.N., and Ramoshebi, L.N. (2005). Bone morphogenetic proteins in craniofacial and periodontal tissue engineering: experimental studies in the non-human primate Papio ursinus. Cytokine Growth Factor Rev 16, 357-368. 10.1016/j.cytogfr.2005.02.006
- Torii, D., Konishi, K., Watanabe, N., Goto, S., and Tsutsui, T. (2015). Cementogenic potential of multipotential mesenchymal stem cells purified from the human periodontal ligament. Odontology 103, 27-35. 10.1007/s10266-013-0145-y.
- Torii, D., Soeno, Y., Fujita, K., Sato, K., Aoba, T., and Taya, Y. (2016). Embryonic tongue morphogenesis in an organ culture model of mouse mandibular arches: blocking Sonic hedgehog signaling leads to microglossia. In Vitro Cell Dev Biol Anim 52, 89-99. 10.1007/s11626-015-9951-6.
- Weiskirchen, R., Meurer, S.K., Gressner, O.A., Herrmann, J., Borkham-Kamphorst, E., and Gressner, A.M. (2009). BMP-7 as antagonist of organ fibrosis. Front Biosci (Landmark Ed) 14, 4992-5012. 10.2741/3583.
- Yamada, S., Tomoeda, M., Ozawa, Y., Yoneda, S., Terashima, Y., Ikezawa, K., Ikegawa, S., Saito, M., Toyosawa, S., and Murakami, S. (2007). PLAP-1/asporin, a novel negative regulator of periodontal ligament mineralization. J Biol Chem 282, 23070-23080. 10.1074/jbc.M611181200.
- Transcriptome analysis has been very well done both in the illustrations and explanations in the manuscript. However, there requires a better resolution in Figures 3C and D.
--> The resolution seems to have dropped a bit in the process of converting the picture file. For better resolution, please check the following picture direct cropped from PPT file. Figure 3 C & D in manuscript was displaced with this picture. Also we send a pdf file for better resolution (Please find this picture in the attached file).
- Figures 4A-C have been well illustrated and interpreted by the author(s) in the manuscript.
- It is recommended that tables 2 and 3 be simplified with their family groups. Meanwhile, each protein's gene description and expression are presented in the data supplement.
--> As you already know, gene ontology has many overlapping definitions, and the genes shown in Tables 2 and 3 of this paper were selected from the genes already overlapping in the ontology of cell adhesion, cell membrane factor, and extracellular matrix factor. Please understand that we cannot accurately subdivide into separate groups again. However, also consider that the same group of genes is rearranged in Table 4.
